# Node-Level Data Valuation on Graphs

**Simone Antonelli** *simone.antonelli@cispa.de*
*CISPA Helmholtz Center for Information Security*

**Aleksandar Bojchevski** *a.bojchevski@uni-koeln.de*
*University of Cologne*

**Reviewed on OpenReview:** *https://openreview.net/forum?id=tNyApIqDSJ*

## Abstract

How much is a node worth? We answer this question using an emerging set of data valuation techniques, where the value of a data point is measured via its marginal contribution when added to the (training) dataset. Data valuation has been primarily studied in the i.i.d. setting, giving rise to methods like influence functions, leave-one-out estimation, data Shapley, and data Banzhaf. We conduct a comprehensive study of data valuation approaches applied to graph-structured models such as graph neural networks in a semi-supervised transductive setting. Since all nodes (labeled and unlabeled) influence both training and inference we construct various scenarios to understand the diverse mechanisms by which nodes can impact learning and final predictions. We show that the resulting node values can be used to identify (positively and negatively) influential nodes, quantify model brittleness, detect poisoned data, and accurately predict counterfactuals.[1]

## 1 Introduction

When we learn from data, a natural question is *how each data point influences learning*. We can study how a given training point affects the learned weights, accuracy, or prediction for a given test point. Beyond offering insights into our models, this question is also practically important, especially for modern machine learning methods using increasingly larger datasets (Zhou et al., 2017). For example, if we gather data from unreliable sources (e.g., the internet) we may want to filter out low-quality instances. When obtaining data from data markets, different providers (e.g., individuals, hospitals) should be equitably compensated for the data they provide. Data values have been used to quantify train-test leakage, find semantically similar examples (Ilyas et al., 2022), detect mislabeled instances (Wang & Jia, 2023), and dataset selection (Engstrom et al., 2024).

A classical answer to the data valuation question is influence functions (Cook & Weisberg, 1982; Koh & Liang, 2017). They approximate the effect of removing a data point on the model performance. From linear regression to deep learning, they have been thoroughly studied in various settings, including their limitations (Basu et al., 2021; Bae et al., 2022). Broadly, data valuation encompasses techniques that try to relate some output of a model with the data it is trained on. While there is growing research that explores different notions of value, the vast majority is focused on supervised learning for i.i.d. data.

We conduct the first comprehensive data valuation study for graph-based models in the semi-supervised transductive setting. Only two recent works have explored data valuation for graphs. Chen et al. (2023) derive a closed-form estimate for the leave-one-out influence of nodes and edges using a simple graph convolution (SGC) network (Wu et al., 2019) as a surrogate model. Chi et al. (2025a) propose the precedence-constrained winter value where nodes are grouped in coalitions based on the graph structure. Our focus is not on proposing new notions, but on rigorously evaluating existing ones. Our results show that both data Banzhaf (Wang & Jia, 2023) and datamodels (Ilyas et al., 2022) significantly outperform other valuation methods, but neither

---

[1]The code is provided at `https://github.com/siantonelli/graph_valuation`.

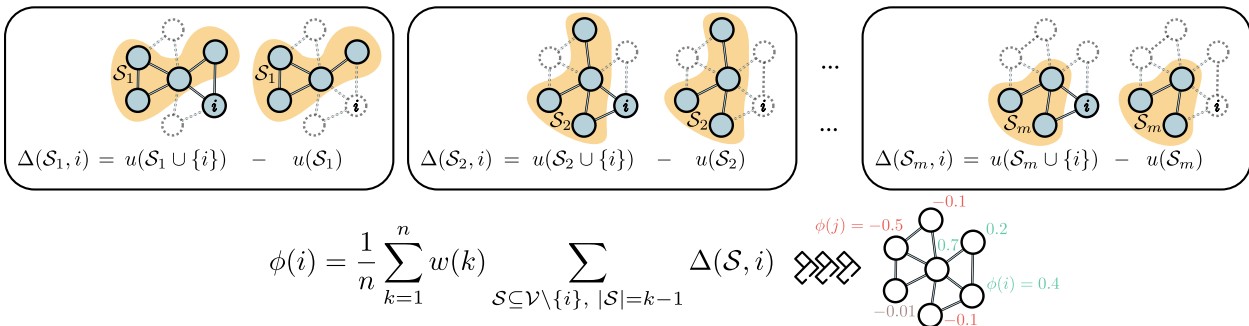

$$\phi(i) = \frac{1}{n}\sum_{k=1}^{n} w(k) \sum_{\mathcal{S}\subseteq\mathcal{V}\setminus\{i\},\ |\mathcal{S}|=k-1} \Delta(\mathcal{S}, i)$$

Figure 1: Data valuation assigns importance to a node based on its marginal contribution to the utility $u$ across various subsets $\mathcal{S}$. When removing the node (and its edges): if model performance decreases, it is considered informative (green); if performance remains unchanged, the node is irrelevant (grey); if performance improves, the node is misleading (red). Weighting schemes $w$ define different data valuation methods.

has been studied for graphs so far. Given the importance and ubiquity of models such as graph neural networks (Ju et al., 2024), we hope our work lays a solid foundation for studying data valuation on graphs.

It is clear that not all data are created equally, e.g. some instances may be noisy or mislabeled. However, even a "high-quality" instance may not be valuable if there are already similar instances in the dataset, indicating that to properly evaluate the worth of an instance we need to carefully consider the appropriate context. Lack of context is why leave-one-out estimation often fails in practice – its focus is on the effect of an instance in isolation. In contrast, recent game-theoretic notions such as data Shapley (Ghorbani & Zou, 2019; Jia et al., 2019b) and other semi-values (Dubey et al., 1981) compute the average marginal contribution of an instance considering all potential subsets of the data. This indeed comes at a significant computational cost, but approximation strategies allow us to compute useful estimates in practice.

The right context is even more important in the graph setting. First, the prediction of a node depends on its neighborhood – nodes are not i.i.d. This suggests that looking at nodes in isolation is likely suboptimal and we should consider different subgraphs. The main idea is illustrated in Fig. 1. To estimate the influence of node $i$ on some utility function $u(\cdot)$ (e.g. the accuracy), we consider the difference in utility $\Delta(\mathcal{S}, i) = u(\mathcal{S}\cup\{i\}) - u(\mathcal{S})$ with and without node $i$ across many subsets $\mathcal{S} \subseteq \mathcal{V}$ of different sizes. Marginal contributions $\Delta(\mathcal{S}, i)$ are merged together using a weighted average to obtain the final node value $\phi(i)$. Different weights recover many of the established data value notions, including data Shapley, beta Shapley, data Banzhaf, and leave-one-out.

Second, all nodes, either labeled and unlabeled, influence both training and inference. We account for this by constructing different variants (see § 3) of each data value notion that aim at disentangling these effects. Unsurprisingly, even though the number of training nodes is a small percentage of all nodes in the sparsely-labeled scenario (Shchur et al., 2018), the top most influential nodes are mostly training nodes. At the same time, we can find unlabeled nodes which have greater influence than many training nodes. This reflects the complex interplay between training and test nodes in graphs.

In addition to influence-based and game-theoretic notions, predictive approaches, like datamodels (Ilyas et al., 2022) and MLPbV (Wu et al., 2024), construct surrogate (proxy) models that can predict the utility function for any subset without the need to train the model from scratch. They allow us to answer counterfactual questions, i.e. predicting the performance on arbitrary unseen subsets of data. Proxy models are trained on subsets of all possible $\{(\mathcal{S}_i, u(\mathcal{S}_i)\}$ pairs, where $\mathcal{S}_i \subseteq \mathcal{V}$. As we discuss in § 2.1, we can trivially turn any set of data values into a predictive (linear) surrogate. Moreover, the resulting data values from predictive and game-theoretic approaches are equivalent for certain configurations (see § 2.2). Thus, this distinction between these types of notions may not be very useful.

Overall, our results show that approaches accounting for subgraphs instead of single-node contribution yield more accurate data values. We demonstrate their usefulness for downstream applications: i) *finding highly influential nodes* – pruning nodes with high positive (negative) influence causes a drop (rise) in performance; ii) *spotting brittle predictions* which depend on a small set of support nodes whose removal causes

misclassification; iii) *detecting poisoned (mislabeled) data*; iv) *estimating counterfactuals* such as predicting performance on an arbitrary subset; and v) *visualizations* that provide further insight about the data.

We thoroughly study the problem of data valuation for graphs, including variants that attribute importance to both labeled and unlabeled nodes. We analyse game-theoretic and predictive notions and show that data-models and data Banzhaf, neither of which was previously considered in the context of graphs, outperform other valuation techniques, including the two proposed graph-specific approaches. Our study was computationally expensive (over 2500 compute hours in total) and storage-intensive (more than 10TB of raw data). Nonetheless, we show that one can obtain accurate estimates even with modest computational resources.

## 2 Data valuation

Broadly, data valuation methods try to relate the presence or absence of an instance in the dataset with the resulting change in some user-defined utility function. Traditionally, this is some measure of model performance such as accuracy, where a high positive (negative) value indicates that including the instance in the data improves (deteriorates) performance. For both game-theoretic and predictive approaches the main idea is to consider different subsets of the training set $\mathcal{D}$ and assess the value of missing training samples according to the change in the utility. We denote with $\phi(i) \in \mathbb{R}$ the value assigned to instance $i$.

**Utility.** Let $u : 2^{|\mathcal{D}|} \to \mathbb{R}$ be a utility function which maps any subsets $\mathcal{S} \subseteq \mathcal{D}$ to a score indicating the usefulness of the subset. For classification tasks, we commonly have $u(\mathcal{S}) = \text{acc}(\mathcal{A}(\mathcal{S}))$ where $\mathcal{A}$ is a learning algorithm that returns a model $f_{\mathcal{S}}$ trained on $\mathcal{S}$, and acc computes the accuracy on a held-out set. Sometimes we are not interested in the overall performance but rather how the training data influences a specific test point. For example, the utility can be the prediction margin for the test point $\boldsymbol{x}_{\text{test}}$. That is $u(\mathcal{S}) = u(\mathcal{S}, \boldsymbol{x}_{\text{test}}) = \text{margin}(f_{\mathcal{S}}, \boldsymbol{x}_{\text{test}}) = f_{\mathcal{S}}(\boldsymbol{x}_{\text{test}})_{y^*} - \max_{y \neq y^*} f_{\mathcal{S}}(\boldsymbol{x}_{\text{test}})_y$ where $f_{\mathcal{S}}(\cdot)_y$ returns the logit-score or probability for label $y$, and $y^*$ is the ground-truth label. Any other function of (the outputs of) $f_{\mathcal{S}}$ is valid.

### 2.1 Categorization of data valuation approaches

**Game-theoretic notions.** These methods consider subsets (coalitions) of the training data, treating each data point as a player in a cooperative game. If $u(\mathcal{D})$ represents the outcome of the game, the goal is to fairly distribute it to each data point according to its contribution. This concept is at the core of Shapley values (Shapley, 1953; Ghorbani & Zou, 2019), a well-known framework for assessing data value via the average marginal contribution of adding a data point to all possible subsets, weighted by the number of permutations in which that data point appears.[2] Let $n = |\mathcal{D}|$. The Shapley value of a data point $i$ is given by

$$\phi_{\text{SHAP}}(i) = \frac{1}{n} \sum_{k=1}^{n} \binom{n-1}{k-1}^{-1} \sum_{\substack{\mathcal{S} \subseteq \mathcal{D} \setminus \{i\} \\ |\mathcal{S}| = k-1}} \left[ u(\mathcal{S} \cup \{i\}) - u(\mathcal{S}) \right] \tag{1}$$

To compute it exactly, we need to train $2^{|\mathcal{D}|}$ models, i.e. one for each possible subset. Since this is prohibitively expensive there are various approximation techniques such as Monte Carlo sampling. Shapley values are popular since they uniquely satisfy four axioms that were argued to be necessary to ensure a fair valuation. These are *symmetry*, *linearity*, *null player*, and *efficiency* (Shapley, 1953).

Kwon & Zou (2022) questions the necessity of the *efficiency* axiom, which requires all the values to sum up to the utility on the original dataset, i.e. $\sum_{i \in \mathcal{D}} \phi(i) = u(\mathcal{D})$. Removing this axiom we get the so called *semivalues* (Dubey et al., 1981) which satisfy the other three axioms. As with any axiomatic approach, these axioms can be debated. For the purposes of this paper, we consider them as given. It turns out that all semivalues can be written in a canonical form

$$\phi_{\text{semivalue}}(i, w) = \frac{1}{n} \sum_{k=1}^{n} w(k) \sum_{\substack{\mathcal{S} \subseteq \mathcal{D} \setminus \{i\} \\ |\mathcal{S}| = k-1}} \left[ u(\mathcal{S} \cup \{i\}) - u(\mathcal{S}) \right] \tag{2}$$

---

[2]Shapley values were first used machine learning for feature attribution – determining which features (e.g. pixels) highly influence the prediction. That is, they served as a post-hoc explainability method. In contrast, Ghorbani & Zou (2019) introduce data Shapley to assess the influence of training points, which is the setting that we consider in this paper (see also § 5).

where $w : [n] \to \mathbb{R}$ is a weight function such that $\sum_{k=1}^{n} \binom{n-1}{k-1} w(k) = n$. We can recover Shapley (`SHAP`) with $w(k) = \binom{n-1}{k-1}^{-1}$. Leave-one-out (`LOO`) measures the difference in utility when removing a single data point from the dataset and can be obtained by choosing $w(k) = n\mathbf{1}[k = n]$. Data Banzhaf (`BANZ`) assigns uniform weights $w(k) = \frac{n}{2^{n-1}}$. Weighted Banzhaf ($\alpha$-`BANZ`) has $w(k) = n\alpha^{k-1}(1 - \alpha)^{n-k}$ weights (Li & Yu, 2023).

**Predictive notions.**   Wang et al. (2021) propose to learn a surrogate model on utilities. Ilyas et al. (2022) introduce the datamodels framework, where $m_{\boldsymbol{\theta}} : \{0, 1\}^{|\mathcal{D}|} \to \mathbb{R}$ is trained such that can directly predict the utility for any given subset, i.e. $m_{\boldsymbol{\theta}}(\mathcal{S}) \approx u(\mathcal{S})$. The weights $\boldsymbol{\theta}$ of the surrogate are learned from $\{(\mathcal{S}_i, u(\mathcal{S}_i))\}$ where $\mathcal{S}_i$ is encoded as a binary vector $\{0, 1\}^{|\mathcal{D}|}$ indicating the presence or absence of an instance. They show that even a simple linear model can approximate the mapping from $\mathcal{S}$ to $u(\mathcal{S})$ well. The data values are given by the weights $\boldsymbol{\theta} \in \mathbb{R}^{|\mathcal{D}|}$ of the linear model, i.e. $\phi_{\texttt{DM}}(i; \boldsymbol{\theta}) = \theta_i$. Wu et al. (2024) propose to instead use a multi-layer perception as a surrogate which can use additional inputs.

**Influence functions.**   The number of possible subsets increases exponentially with the size of the dataset. Moreover, for each subset, we need to train the entire model from scratch. If the dataset is large even computing the leave-one-out error which requires "only" $n = |\mathcal{D}|$ evaluations can be too expensive. In contrast, influence functions (Cook & Weisberg, 1982) aim to approximately compute the change in the model output (or model weights) when removing a training sample. They rely on first and second-order information (gradient, Hessian) to obtain estimates without model retraining. Nevertheless, influence functions perform poorly in non-convex settings as in deep neural networks (Basu et al., 2021; Bae et al., 2022). At best, they perform as well as leave-one-out errors which are suboptimal since they consider instances in isolation. Chen et al. (2023) derive different data values using influence functions for graph neural networks.

**From game-theoretic to predictive.**   Any game-theoretic approach can be made predictive by assuming that the values are the weights of an implicit linear model (without a bias term). This implies that the predicted value of any unseen subset equals the sum of the data values of the nodes in the subset, i.e. $u(\mathcal{S}) \approx \sum_{i \in \mathcal{S}} \phi(i)$ for any $\phi(i)$. This is a reasonable assumption given the linearity axiom of semivalues, and it's explicitly made by datamodel values (Ilyas et al., 2022).

## 2.2   Properties of data valuation approaches

**Connections between different values.**   To compute both `DM` and $\alpha$-`BANZ` we sample subsets where each node is included with probability $\alpha$. If $\alpha = 0.5$ and we use no regularization when fitting $m_{\boldsymbol{\theta}}$ the datamodel values are equivalent to Banzhaf values since the Banzhaf values are the best linear approximation to $u$ in terms of least square loss (Wang & Jia, 2023; Hammer & Holzman, 1992). Lin et al. (2022) prove that if we sample $\alpha \sim \mathrm{Unif}(0, 1)$ rather than having it fixed, then the optimal weights of a certain regularized linear model converge relatively quickly to the Shapley values as we increase the number of samples.

**Limitations of data values.**   The utility function can be viewed as a vector in $\mathbb{R}^{2^n}$ which is transformed into a vector of data values $\boldsymbol{\phi} \in \mathbb{R}^n$. This mapping is not injective, so there are different utility functions that yield identical values. A natural question is which utility functions are well approximated by data values. Wang et al. (2024) show that a sufficient condition is for the utility to be a monotonically transformed modular function. In general, data values may be no better than a random baseline for dataset selection. They also generalize the result from Saunshi et al. (2022) which states that the residual error of the unregularized linear approximation equals to the sum of all $u$'s Fourier coefficients of order 2 or higher. Nonetheless, data values work well in practice. This is remarkable since they approximate $u(\mathcal{S})$ which involves training of a model on $\mathcal{S}$ from scratch (often a complicated neural network) and then computing some function of the model outputs.

**Robustness and efficiency.**   In our context the utility function is stochastic due to randomness in the training process, making the influence of a data point a random variable with an expectation and variance (Nguyen et al., 2024). Thus, using a single sample of $u(\mathcal{S})$ may be unreliable. Wang & Jia (2023) define the safety margin – the largest noise a semivalue can tolerate without altering the ranking of any distinguishable pair of data points. They prove that data Banzhaf achieves the largest safety margin among all values. This follows from the uniform weights since $\phi_{\texttt{BANZ}}(i) = \mathbb{E}_{\mathcal{S} \sim \mathrm{Unif}(2^{n \setminus i})}[u(\mathcal{S} \cup \{i\}) - u(\mathcal{S})]$. The standard Monte Carlo estimator samples $\mathcal{S}$ from $\mathrm{Unif}(2^{n \setminus i})$ but must be repeated $n$ times, once for each $i$. Wang & Jia (2023) propose an alternative estimator where given samples $\mathcal{M} = \{\mathcal{S}_1, \ldots, \mathcal{S}_m\}$ i.i.d. from $\mathrm{Unif}(2^n)$ it computes

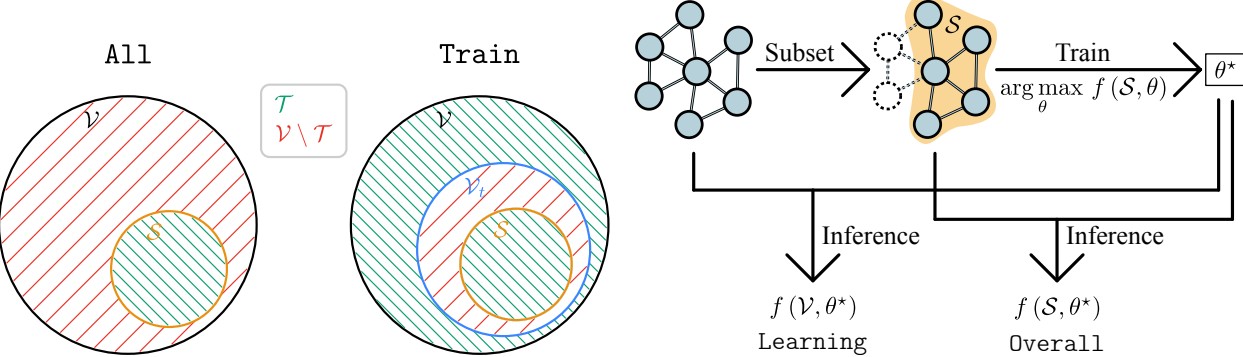

Figure 2: Venn diagram of node subsets for their induced subgraphs. Downward hatches from left to right (green) indicate the final subset $\mathcal{T}$.

Figure 3: Pipeline for the `learning` vs. `overall` signals: train on subset $\mathcal{S}$, then measure utility either on $\mathcal{V}$ or $\mathcal{S}$, respectively. Here, $\mathcal{T} = \mathcal{S}$ (`all` setting).

$\phi_{\texttt{BANZ}}(i) = \frac{1}{|\mathcal{S}_{\in i}|} \sum_{\mathcal{S} \in \mathcal{S}_{\in i}} u(\mathcal{S}) - \frac{1}{|\mathcal{S}_{\notin i}|} \sum_{\mathcal{S} \in \mathcal{S}_{\notin i}} u(\mathcal{S})$ as the difference of the average utility of the sets $\mathcal{S}_{\in i} \subseteq \mathcal{M}$ that contain $i$, minus the sets $\mathcal{S}_{\notin i} \subseteq \mathcal{M}$ that do not contain it. Now, maximal sample reuse (MSR) is achieved since all evaluations of $u(\cdot)$ are used in the estimation of all $\phi_{\texttt{BANZ}}(i)$ values. The existence of an efficient MSR estimator is a unique to the Banzhaf value among all semivalues. Liu et al. (2024) also show that the latter can be approximated with a linear regression formulation. Nonetheless, we apply the same idea of Li & Yu (2023) and use $\alpha$-`BANZ` where instances in the subsets are sampled with a fixed probability $\alpha$ (rather than uniform). In our experiments, $\alpha$-`BANZ` outperforms `BANZ` and shows strong results across all settings.

## 3 Data valuation on graphs

**Semi-supervised setting.** Consider a graph $\mathcal{G} = (\mathcal{V}, \mathcal{E})$ where $\mathcal{V}$ is the set of nodes and $\mathcal{E}$ is the set of edges. Let $\boldsymbol{X} \in \mathbb{R}^{|\mathcal{V}| \times d}$ be the matrix of $d$-dimensional node features, $\boldsymbol{A} \in \{0, 1\}^{|\mathcal{V}| \times |\mathcal{V}|}$ be the adjacency matrix and $\boldsymbol{y} \in \mathbb{R}^{|\mathcal{V}|}$ be the vector of labels. Let $\mathcal{V}_\ell$ and $\mathcal{V}_u$ be the subsets of labeled and unlabeled nodes respectively, and let $\mathcal{V}_\ell$ be further split into training nodes $\mathcal{V}_t$ and validation nodes $\mathcal{V}_v$. Usually, $\mathcal{V}_\ell$ is a small subset of nodes while the majority is in $\mathcal{V}_u$ (Shchur et al., 2018). In transductive semi-supervised learning setting, the model is exposed to both $\mathcal{V}_\ell$ and $\mathcal{V}_u$ during the training, such that it can propagate label information from the few labeled nodes to the many unlabeled nodes leveraging the underlying graph connectivity and node features.

**Graph Neural Networks (GNNs).** In each layer $k$ of a GNN, the hidden representation $\boldsymbol{h}_v^{(k)}$ of a node $v$ is the result of aggregating information from its neighborhood $\mathcal{N}(v)$ and its own representation from the previous layer. Many popular GNNs can be succinctly written in matrix notation as $\boldsymbol{H}^{(k)} = \sigma\left(\boldsymbol{S} \boldsymbol{H}^{(k-1)} \boldsymbol{W}^{(k)}\right)$, where $\sigma$ is the non-linearity, $\boldsymbol{W}^{(k)}$ are trainable parameters, $\boldsymbol{S}$ is the spatial graph convolution operator and $\boldsymbol{H}^{(0)} = \boldsymbol{X}$. For example, in GCN (Kipf & Welling, 2017) we use the degree normalized adjacency matrix $\boldsymbol{S} = \tilde{\boldsymbol{D}}^{-\frac{1}{2}} \tilde{\boldsymbol{A}} \tilde{\boldsymbol{D}}^{-\frac{1}{2}}$, where $\tilde{\boldsymbol{A}} = \boldsymbol{A} + \boldsymbol{I}_{|\mathcal{V}|}$ and $\tilde{\boldsymbol{D}}_{ii} = \sum_j \tilde{\boldsymbol{A}}_{ij}$ is the degree matrix. SGC (Wu et al., 2019) uses the same architecture as GCN without the non-linearity resulting in a model that is linear w.r.t. the weights.

**Training nodes vs. All nodes.** In the transductive setting, all nodes influence learning. The training nodes do so directly since the loss is computed using their ground-truth labels as supervision signal. The remaining nodes do so indirectly since they influence the hidden representation of the training nodes via the graph structure. This means that in addition to investigating the value of the training nodes $\mathcal{V}_t$ as in the standard i.i.d. setting, we can also compute the value of all nodes $\mathcal{V}$. We refer to these two settings as `train` and `all` respectively. Let $\mathcal{D}$ be the set of considered nodes, namely either $\mathcal{D} = \mathcal{V}_t$ or $\mathcal{D} = \mathcal{V}$ for the `train` or `all` setting respectively. Recall that to compute data values we consider different subsets $\mathcal{S} \subseteq \mathcal{D}$. We first remove the nodes that are in $\mathcal{D}$ but not in $\mathcal{S}$, and then we train a GNN on the graph induced by the remaining nodes $\mathcal{T} = \mathcal{V} \setminus (\mathcal{D} \setminus \mathcal{S})$. In the `train` setting, $\mathcal{T} = \mathcal{V} \setminus (\mathcal{V}_t \setminus \mathcal{S})$ and in the `all` setting $\mathcal{T} = \mathcal{V} \setminus (\mathcal{V} \setminus \mathcal{S}) = \mathcal{S}$.

Fig. 2 shows Venn diagram representations of the considered subgraphs in the two settings. Since all nodes influence learning we focus on the `all` setting in the main paper. However, `train` results are in § C.4.

**Learning signal vs. Overall signal.** After training, we need to evaluate our utility function. We consider the *accuracy* to measure the influence on the final performance, and the *prediction margin* to measure influence w.r.t. any individual node (see § 2). Here, we assume access to ground-truth test labels, since our goal is to understand how nodes influence learning for different models. This is standard in the data valuation literature. In practical scenarios, one can use validation labels instead. Importantly, after training on the graph induced by $\mathcal{T}$ we can compute the utility in two different ways which provide different and complementary insights. In particular, we can compute the utility on the same induced subgraph $\mathcal{T}$, which captures the `overall` influence of a node since removed nodes are not present during inference. Alternatively, we can compute the utility on the whole graph $\mathcal{V}$ – that is we bring back the removed nodes $\mathcal{V} \backslash \mathcal{T}$ – which only captures their influence during learning. These `learning` data values measure the training signal provided by a node, while the `overall` data values measure the influence during training and inference. This is in contrast to the i.i.d. setting where we can only measure the train signal provided by an instance. Fig. 3 depicts the pipelines for computing the values in the two described settings for the `all` setting.

## 4 Experimental results and analysis

**Setting.** As representative game-theoretic notions, we consider three well-known semivalues: leave-one-out (`LOO`), data Shapley (`SHAP`), and weighted Banzhaf values ($\alpha$-`BANZ`), where $\alpha = 0.5$ recovers the original Banzhaf value. As a representative predictive notion, we consider datamodels (`DM`). For approaches based on MSR we describe the procedure to sample subsets in § B.2. Finally, we consider the precedence-constrained winter value (`PCW`) as the only data value custom-designed for graphs since the method by Chen et al. (2023) approximates `LOO`, which we already cover. We introduce a sampling improvement to `PCW` which we call `PCWP` (see § B.2.2 for details). We include two more baselines, namely `DEG` where a node value corresponds to its degree, and `RND` which randomly assigns node values.

We evaluate methods on the largest connected component (LCC) of different citation graphs (`Citeseer`, `CoraML`, `PubMed`, and `CoPhysics`) and co-purchase graphs (`Photo` and `Computers`). We also analyze heterophilic datasets in § C.1. We use SGC, GCN, and GAT models. Unless specified otherwise, we present results on SGC. See § B for datasets, models, and runtime details. We assume that a node's value depends on both its features and its edges, since we completely remove any node $v \notin \mathcal{S}$. Disentangling features from structure is an interesting direction for future research.

**Key findings.** We summarize our results in five key findings, each supported by one or more experiments.

> **I.** Nodes exert positive influence within their class and negative influence across classes.

We study how every node influences every other node using prediction margins as utilities. We obtain a matrix of values $\mathbf{\Phi} \in \mathbb{R}^{|\mathcal{V}| \times |\mathcal{D}|}$, where a column $j$ of $\mathbf{\Phi}$ shows how node $j$ influences other nodes. A positive value $\mathbf{\Phi}[i, j]$ means that adding node $j$ to the graph increases the margin of node $i$, while a negative value means that the margin decreases. On Fig. 4, we show a heatmap of the entire matrix $\mathbf{\Phi}_{\text{DM}}$ (`overall` signal, `all` setting) using GCN to compute utilities on `CoraML`. We also annotate training nodes and highly influential nodes (outliers) with bigger markers. Since rows and columns are sorted by class, cluster size and degree, the block-diagonal pattern with positive diagonal and negative off-diagonal entries, leads to the conclusion stated in Finding I. The results are consistent for different methods, and we observe no qualitative difference between `learning` and `overall` values (§ A.1).

In the average per-class influence plot (Fig. 4, top right), we summarize $\mathbf{\Phi}_{\text{DM}}$ by averaging node influence for every class pair. Influence is positive within a class and negative across classes, and adding same-class nodes widens this margin. Cluster size strongly correlates with within-class influence, i.e. adding a node to a large community matters less because many others play the same role, and this holds across all datasets (see § A.1). Influence is asymmetric, e.g. *Theory* nodes affect the *Rule* class more than the reverse, and the smallest communities show the strongest negative cross-class influence. These patterns warrant further study.

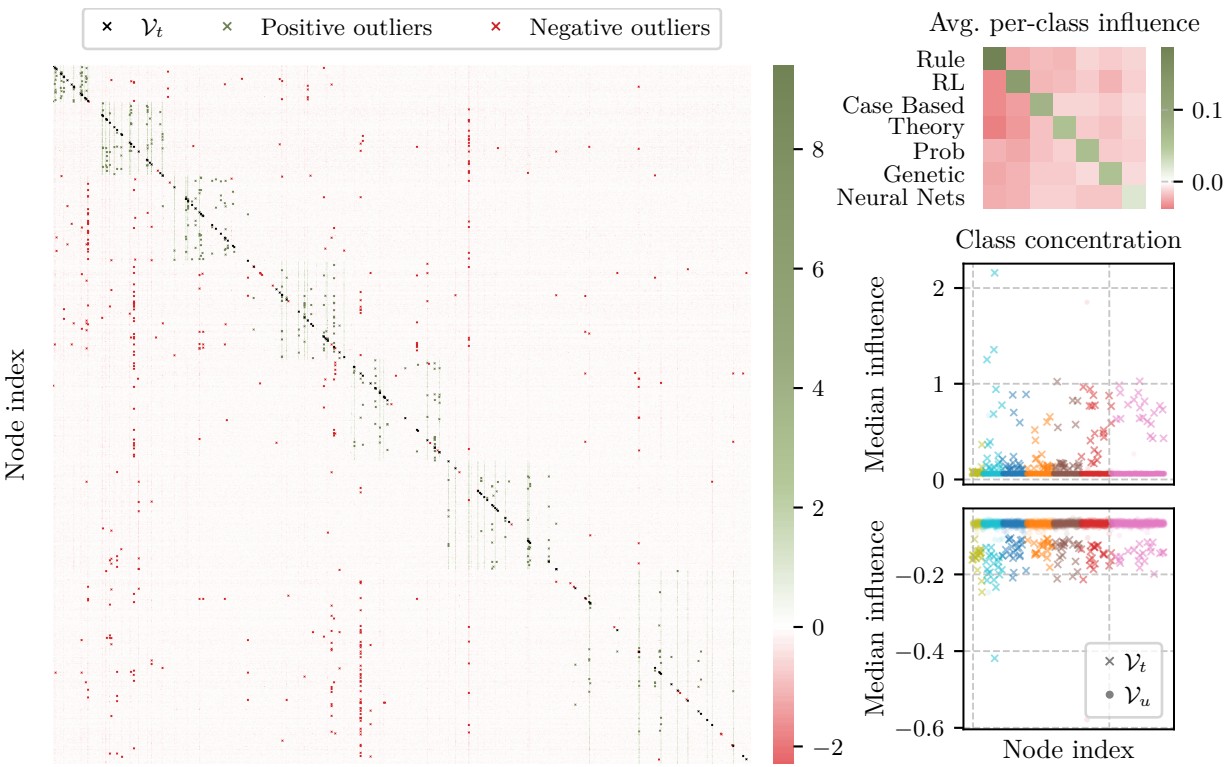

Figure 4: (Left) Matrix of $\mathbf{\Phi}_{\text{DM}}$ on `CoraML` from GCN utilities, where each cell shows the positive (green) or negative (red) effect of adding node $j$ on node $i$'s margin. (Top right) Per-class averages from $\mathbf{\Phi}_{\text{DM}}$ shows positive intra-class and negative inter-class effects. (Bottom right) Median positive/negative influences (colored by class) from $\mathbf{\Phi}_{\text{DM}}$ reveal that a few (training) nodes dominate (vertical stripes in $\mathbf{\Phi}_{\text{DM}}$).

**II.** Influence is concentrated and synchronized within each class.

**Concentrated.** A small subset of nodes (mostly training nodes) is highly influential. This can be easily seen from the $\mathbf{\Phi}_{\text{DM}}$ heatmap on Fig. 4 where a few columns have large positive and negative values across many rows, visible as vertical strips. In the class concentration subplots on Fig. 4 (right), we show the median (per column) influence of a node and split them in positive in the top plot and negative in the bottom plot. We color the nodes by class and mark them as training (crosses) and non-training (circles). The results confirm that most of the influence (either positive or negative) *concentrates* among training nodes and a few of those have a high median value highlighting it is influential for many nodes. Since we sorted $\mathbf{\Phi}_{\text{DM}}$ by class, cluster size and degree, we can actually see how the degree plays an important role for the positive influential nodes while less important for the negative ones. In § C.9, we study how this behaviour is linked to graph connectivity.

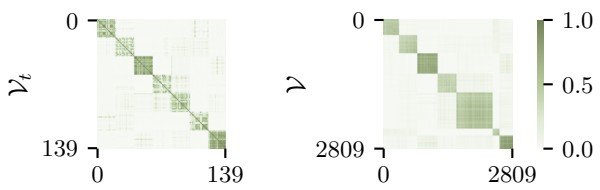

Figure 5: Pairwise $R^2$ for `CoraML`'s $\mathbf{\Phi}_{\text{DM}}$. Column-wise $R^2$ (left) shows same-class training nodes ($\mathcal{V}_t$) affect others similarly; Row-wise $R^2$ (right), shows same-class nodes ($\mathcal{V}$) also share influence.

**Synchronized.** We additionally see high correlation between many pairs of columns. To study this further, we compute the pairwise $R^2$ coefficient of entries in $\mathbf{\Phi}_{\text{DM}}$ (either rows or columns) in the `train` setting `learning` signal on `CoraML`. From the block-diagonal structure on Fig. 5 (left) we see that any pair of

training nodes $i$ and $j$ from the same class *influence* all other nodes in the same way since the $R^2$ score between $\mathbf{\Phi}[:, i]$ and $\mathbf{\Phi}[:, j]$ is high. Similarly, on Fig. 5 (right) we also see that any pair of same-class nodes *are being influenced* in the same way by training nodes as show by high $R^2$ score between $\mathbf{\Phi}[i, :]$ and $\mathbf{\Phi}[j, :]$. Since nodes within each class both exert and receive influence in similar ways, we refer to this effect as synchronization. We observe analogous behavior in the `all` setting and `overall` signal (see § A.2).

---

**III.**   Training nodes are most influential, but a few test nodes stand out.

---

We look at the most positively and negatively important nodes in the two-hop neighbors (to match the receptive field of a two-layer GNN) of a selected test node according to `CoraML`'s $\mathbf{\Phi}_{\text{DM}}$ encoding the `learning` signal. In Fig. 6, we generate a breadth-first tree from a given test node. We color nodes by their classes, size both nodes and edges by the absolute value of their influence w.r.t. the root, and shape nodes as circles if they are training nodes or squares otherwise. Edges are colored green if they positively influence the root node and red otherwise. We see that the most important (positive) nodes are usually training ones (large circles). However, we can find important test nodes (large squares) that are more relevant than some training nodes. These are usually test nodes connected to training nodes from the same class. The class concentration plots in Fig. 4 also confirm this insight (compare crosses to dots). This emphasizes the importance of looking at both training and unlabeled nodes within the graph domain. In § A.4, we rank nodes according to their values in different settings and show how approaches rank non-training nodes vs. training ones. In § A.3, we compare ranks from the `learning` and `overall` signals.

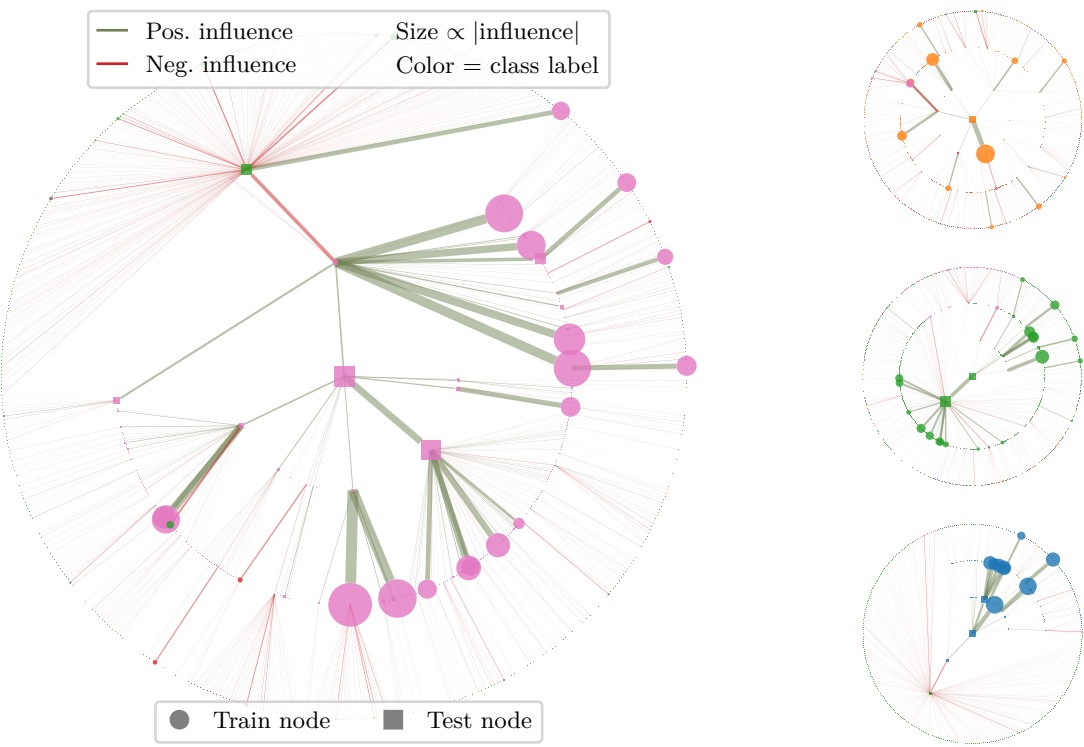

Figure 6: Two-hop breadth-first trees rooted at selected *test* nodes (center of each radial graph) from `CoraML`. Nodes are class-colored, and edges are colored by sign (green for positive, red for negative). Both nodes and edges are sized based on $\mathbf{\Phi}_{\text{DM}}$ influence values. Training nodes (circles) are typically most influential, but some test nodes (squares) stand out, often when highly connected to same-class training nodes.

**IV.** Data values are excellent at detecting brittle and poisoned nodes.

**Brittle.** The support of a node $v$ is the minimum set of nodes such that if removed, the node $v$ is misclassified. This concept was introduced by Ilyas et al. (2022) for images. Using $\mathbf{\Phi}[v, :]$ we can estimate nodes that have high influence on $v$, which in turn allows us to estimate the support. We follow Alg. 1 from Ilyas et al. (2022), using different approaches to estimate the top-$k$ nodes and the support in the `all` setting. On Fig. 7 (left) we see that, regardless of the considered signal (`learning` at the top of the right-hand column, and `overall` at the bottom) most nodes are brittle – more than half of the nodes (y-axis) have a support size of 15 or less (x-axis), and by just removing them we can misclassify more than 50% of nodes. Computing the exact support is intractable and each data value gives us an upper bound. Choosing the best upper bound among all values we arrive at the best estimate marked with a black dash line (ensemble guess). Again we see that DM and $\alpha$-BANZ are closest to this estimate, and are the only two approaches reaching a reasonable estimate in the `overall` setting where all the other fail.

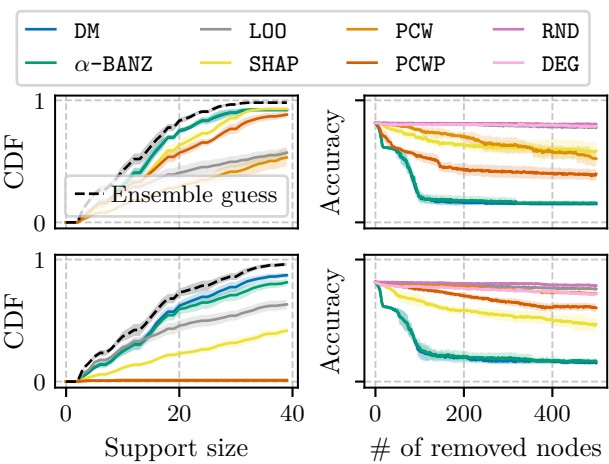

Figure 7: Sampling reuse approaches (`DM` and $\alpha$-`BANZ`) show more consistent performance in predicting supports (left) and estimating values (right). Predicting supports uses `CoraML`'s $\mathbf{\Phi}$ with an SGC model, while node removal uses a GAT model.

In § C.2 we include results for the `train` setting as well, where we can draw similar conclusions.

**Poisoned.** $\mathbf{\Phi}[v, :]$ can also be used to spot corrupted instances. When gathering data from external sources, some instances may be corrupted intentionally (poisoning) or unintentionally (mislabeling). The rank of node values can help detect such instances. Namely, a high data value of a labeled node for itself (self-importance) indicates memorization. When a training node $v$ has a wrong label and $v \in \mathcal{S}$ the model must memorize it to achieve low training loss. If $v \notin \mathcal{S}$ then the model's prediction does not match the wrong label. This leads to a large difference $u(\mathcal{S} \cup \{v\}) - u(\mathcal{S})$, and thus a large value. To test this, we poison 10% of the training data and compute the node values using the margins of the nodes themselves as the utility. We consider the `learning` signals in the `train` setting. In Fig. 9, we show the percentage of poisoned data for `CoraML` appearing in top-$k$ ranked nodes as $k$ increases, with `DM` and $\alpha$-`BANZ` consistently detecting poisoned data better than the others. We provide details in § C.3.

**V.** Sample reuse is a critical feature for data valuation approaches to achieve good performance.

To evaluate the quality of node values, we can again use their rank. Nodes with high positive value should have positive influence on the model, while nodes with high negative value should have the opposite effect. A standard analysis is to then remove (or add) nodes according to the values' rank and observe the change in model's final performance (test accuracy). In Fig. 7 (right), a GAT model computes `learning` (top) and `overall` (bottom) signals, using test margins as utility on `CoraML` (see § B.1 for details on deriving ranking from margins), and rank nodes for removal based on the values. Fig. 8 shows analogous removal results across datasets using a GCN in the `all` and `learning` signal setting, using test accuracy as utility. We expect a steep decline in the performance as we remove the first most important nodes and then a plateau when removing non-influential nodes. We stop after removing 500 nodes since we are only interested in the initial performance drop. Overall, `DM` and $\alpha$-`BANZ` consistently show strong results and have the steepest decline in performance. `PCWP` performs worse, but better than `PCW` and `SHAP`. We omit `DM` on some of the large datasets due to memory issues. `LOO` and `SHAP` converge to the simple `RND` and `DEG` baselines on larger graphs. In contrast, `PCWP` seems to work better on larger graphs (e.g. `PubMed` and `CoPhysics`) but still lags behind `DM` and $\alpha$-`BANZ`. However, the gap between the methods is emphasized in the `overall` setting. Here the valuation methods have "less information" since if node $i \notin \mathcal{S}$ the node is not available during training or inference. In § C, we report additional experiments in-

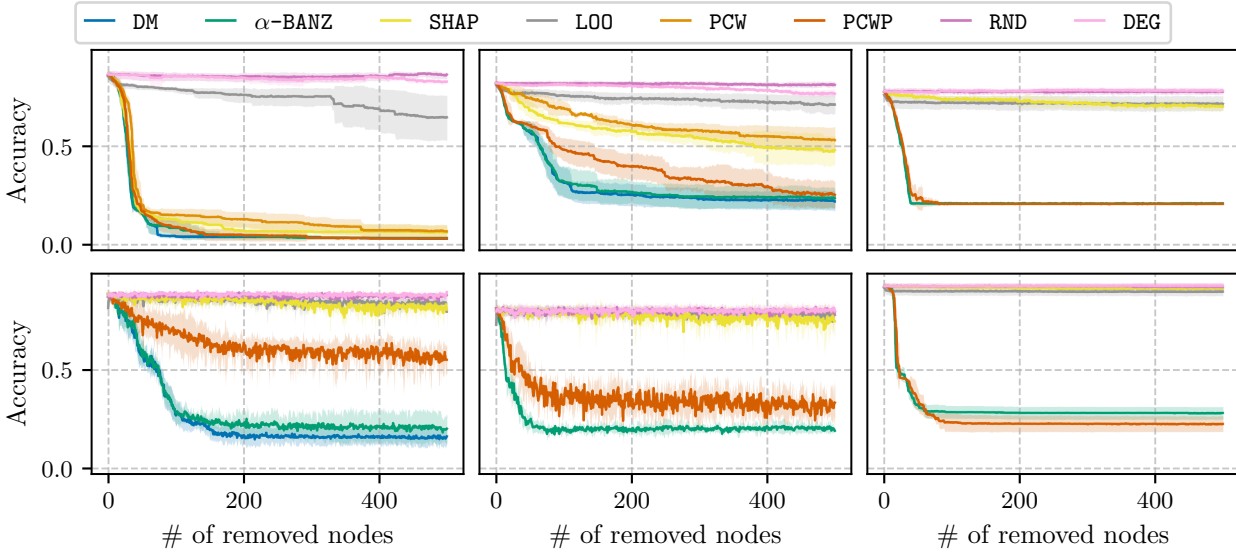

Figure 8: Most influential node removal according to $\mathbf{\Phi}$ computed using GCN: `Citeseer`, `CoraML` and `PubMed` (first row), and `Photo`, `Computers` and `CoPhysics` (second row). Pruning high-value nodes leads to a performance (test accuracy) drop, revealing the quality of estimated values. `DM` and $\alpha$-`BANZ` consistently show the steepest declines across datasets.

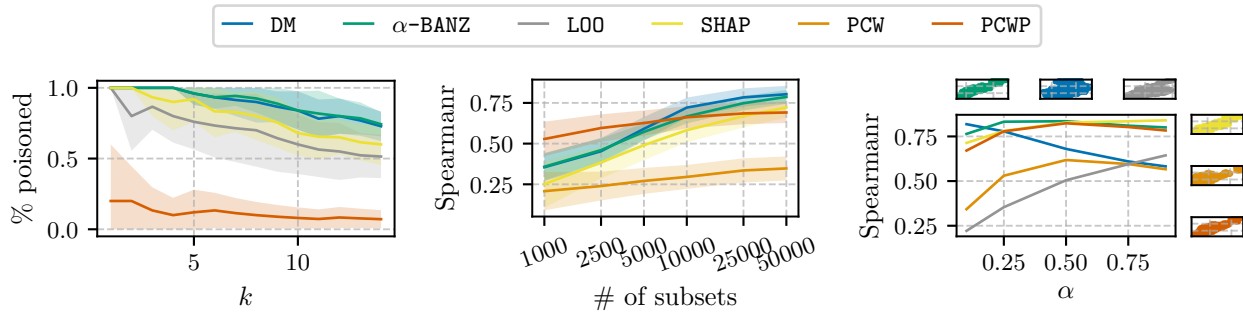

Figure 9: Ratio of poisoned nodes among top-$k$ ranks on `CoraML`, with `DM` and $\alpha$-`BANZ` identifying most of them.

Figure 10: An increasing number of subsets leads to better predictive performance (LDS). Experiment on `CoraML`'s $\mathbf{\Phi}$.

Figure 11: LDS on counterfactual subsets. Insets show predicted vs. true utility correlation at $\alpha = 0.5$. Experiment on `CoraML`'s $\mathbf{\Phi}$.

cluding the `train` setting § C.4, as well as node addition experiment § C.5. The conclusions are similar. Given GNNs' sensitivity to train/val/test splits Shchur et al. (2018), we study value stability across 10 splits in § C.6.

Park et al. (2023) introduce the *linear datamodeling score* (LDS) defined as the Spearman correlation between the predicted utility and the true utility. The idea is that a good (predictive) valuation method should accurately approximate the utility on held-out subsets. Fig. 10 shows the improvement in the LDS as increasing the number of subsets when the held-out set comes from the same distribution (same $\alpha = 0.1$ for both computing and evaluating the data values) for `DM` and $\alpha$-`BANZ`. Fig. 11 shows instead the LDS performance when the distribution of the held-out dataset changes while keeping $\alpha = 0.1$ fixed for computing the data values. From the former, we can see how from already just 10 000 subsets `DM` and $\alpha$-`BANZ` achieve reasonable results. From the latter instead, $\alpha$-`BANZ` generalizes better across the counterfactual sets, followed by `SHAP` and `PCWP`. On the other hand, `DM` becomes worse for a larger distribution shift (larger $\alpha$).

Generally, we notice consistent good performance in approaches taking advantage of the sampling reuse (Fig. 7 to Fig. 10). The data value of an instance is updated leveraging information from all the subsets and not only from the ones where the node is included. Indeed, both `DM` and $\alpha$`-BANZ` take advantage of the sampling reuse trick (we study further the limitations of game-theoretic approaches in the context of graphs in § C.8).

**Discussion and outlook.** Finding I confirms that nodes tend to reinforce intra-class predictions while dampen inter-class ones. The magnitude varies with community size: smaller groups exert stronger cross-class influence, whereas larger ones are redundant, reducing per-node impact. This is because when many nodes carry similar information, their contribution overlap and individual's influence becomes less significant. We find this phenomenon equally compelling for i.i.d. settings and leave its exploration to future work.

Interestingly, Finding II shows that a handful of training nodes dominate the valuation and are synchronized. This *influence homogeneity* suggests opportunities for potential pruning, since we can remove nodes that are simultaneously unimportant for many others, but also vulnerabilities, since a few nodes have a large influence over many test nodes. In the i.i.d. setting, training and test data are presented to the model in separate phases. In the transductive semi-supervised setting, this separation does not occur, and Finding III shows that certain test nodes can exert substantial influence. Understanding what makes these nodes influential, such as by analyzing their neighbor structures, remains an open research direction. With Finding IV, we explore two applications for node values that have also been studied in the i.i.d. context. A natural next step is to identify novel, inherently graph-specific applications for node values.

Finally, Finding V reveals that sample reuse is key to accurate data valuation on graphs as well – Li & Yu (2024) observed this for the i.i.d. setting. Indeed, methods like `DM` and $\alpha$`-BANZ` outperform custom graph approaches (e.g., `PCWP`) by efficiently repurposing subset evaluations, even without neighborhood constraints. This hints at structure-aware MSR techniques benefiting from both adaptive subgraph sampling and statistical sampling reuse. Moreover, with the same subset-evaluation budget, MSR attains higher valuation accuracy, easing scaling limits on large graphs. In § C.7 we show that values learned with a simpler (faster) graph model transfer well to a more complex one, offering another route around computational bottlenecks.

Together, these findings position node-level data valuation not just as a diagnostic tool but as a lens to rethink GNN design: mitigating brittleness via value-aware robust training, exploiting synchronization for efficient learning, and redefining the role of unlabeled data. Bridging game-theoretic principles with graphs, we hope to spark research in this direction with the goal of developing more interpretable, efficient, and robust GNNs.

## 5 Related work

In the literature, data valuation has been adopted mainly in an i.i.d. setting to study relations between training data and changes in the model's utility or to support prediction's explanations at inference time. This section presents an overview of the works that are closest to ours.

**Data valuation in the i.i.d. setting.** The seminal works by Ghorbani & Zou (2019); Jia et al. (2019b) extends Shapley value from feature-level to data point granularity to assess training data importance. As this may result in a computationally expensive procedure, Jia et al. (2021; 2019a) compare the utility of different data attribution methods and propose a fast estimator for Shapley values based on a k-nearest neighbors surrogate for which the exact Shapley values can be efficiently computed. A more recent research direction uses linear surrogates to learn a mapping from a subset of the training set to the model's utility (Ilyas et al., 2022; Wang et al., 2021). The surrogate is then used to estimate the utility from newly sampled subsets and the surrogate weights represent the data values. As the number of possible training subsets is exponential w.r.t. the number of training data points, the learning of the surrogate may be time-consuming. In a follow-up work, Park et al. (2023) employ Taylor approximation to linearize a considered deep neural network such that the one-step Newton approximation can be applied in a non-linear setting. They show that in such a way, just a few training subsets are needed to attribute importance to training data accurately. Wang & Jia (2023) introduce the concept of data Banzhaf for assessing the values of training data and showing its robustness in differentiating data. Except for data Shapley, these approaches have not yet been applied to the graph domain. In our work, we incorporate and compare all these approaches to investigate their behaviour when structural information comes into play. Further, Sundararajan et al. (2020) introduce the Shapley-Taylor

interaction index, a generalization of the Shapley value to feature interactions of bounded order. Following this work, Tsai et al. (2023) propose Faith-Shap, a variant satisfying the standard Shapley axioms via a least-squares formulation, while Fumagalli et al. (2023) develop SHAP-IQ, a unified estimator applicable to any cardinal interaction index. Also, Li & Yu (2024) propose an estimator based on maximum sample reuse to approximate any probabilistic value, notably improving convergence for Shapley-like attributions. Although they address the computational bottlenecks of subset sampling, their evaluation is limited to feature attribution. We leave the investigation of interactive indices in the graph-data valuation setting as future work. Most recently, Liu et al. (2024) introduced Kernel Banzhaf, which frames Banzhaf value estimation as a linear least-squares regression problem with a pair subset sampling procedure that considers the complement of each subset, improving the value estimation.

**Data valuation for graph-structured data.** Graph data valuation relating data to model predictions is still under investigation. To our knowledge, only two works try to tackle this direction. Chen et al. (2023) adapt influence functions to a graph-structured model to approximate the leave-one-out training. However, they consider only training nodes in a transductive setting, without capturing the information coming from explicitly removing unlabeled nodes. We instead consider a more comprehensive picture by studying the transductive scenario, looking at the influence of all nodes, whether labeled or not. Instead, Chi et al. (2025a) propose a new coalition sampling based on the Winter value that considers the graph's structure when creating a permutation of nodes to process and estimate each node's contribution. However, this work focuses on the inductive scenario where unlabeled nodes do not play a role during the model training.

**Graph data valuation at inference time.** When applied at inference time, a data valuation approach assesses the most important input patterns that mostly influence the final model's prediction (feature or node/edge/subgraph attribution). Particularly for graphs, this translates into highlighting the graph structure or node features that cause the prediction for the input. For instance, Duval & Malliaros (2021) compute graph structure and node feature explanations for a single example by constructing a surrogate model on a perturbed graph and computing Shapley values as explanations. Akkas & Azad (2024) explain predictions by computing Shapley values edge-wise and outputting the subgraph with edges from the top-k values. Chhablani et al. (2024) use Banzhaf value in combination with thresholded utility functions on edges to provide counterfactual explanations. Bui et al. (2024) introduce the Myerson-Taylor index which includes structure information inside the Shapley value to identify important motifs for the prediction. Finally, Chi et al. (2025b) introduce a structure-aware Shapley value utility by leveraging the precedence-constrained sampling from `PCW` (Chi et al., 2025a), adapted to the inference setting, and combines it with transferable feature extraction and a Shapley-guided optimization procedure tailored to structure-aware utility learning. Differently from the mentioned works, we employ data valuation to discover relations between nodes involved in the training and the final output of the model. Namely, we do not treat the GNN as a fixed black-box function.

## 6   Conclusion and further research

We present the first extensive study of data valuation methods for graph-structured data. We introduce different data valuation scenarios, and apply state-of-the-art data valuation approaches that have not yet been investigated in the context of graphs. Our results demonstrate that these methods significantly outperform the latest efforts in graph data valuation across multiple applications. We also show how different utility functions can enable several applications. We summarize our results in five key findings that emphasize properties of different methods. Due to the need to train numerous models on different subsets, our study was computationally intensive, and more effort is needed to develop more efficient data valuation methods for node values (both in terms of resources and time efficiency) in particular for large graphs. Future directions also include exploring alternative proxies for datamodels that account for graph structures and a deeper investigation into settings where nodes are simply unlabeled rather than removed.

## Acknowledgments

This work used the RAMSES high-performance computing cluster at the University of Cologne's ITCC, whose support is gratefully acknowledged.

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

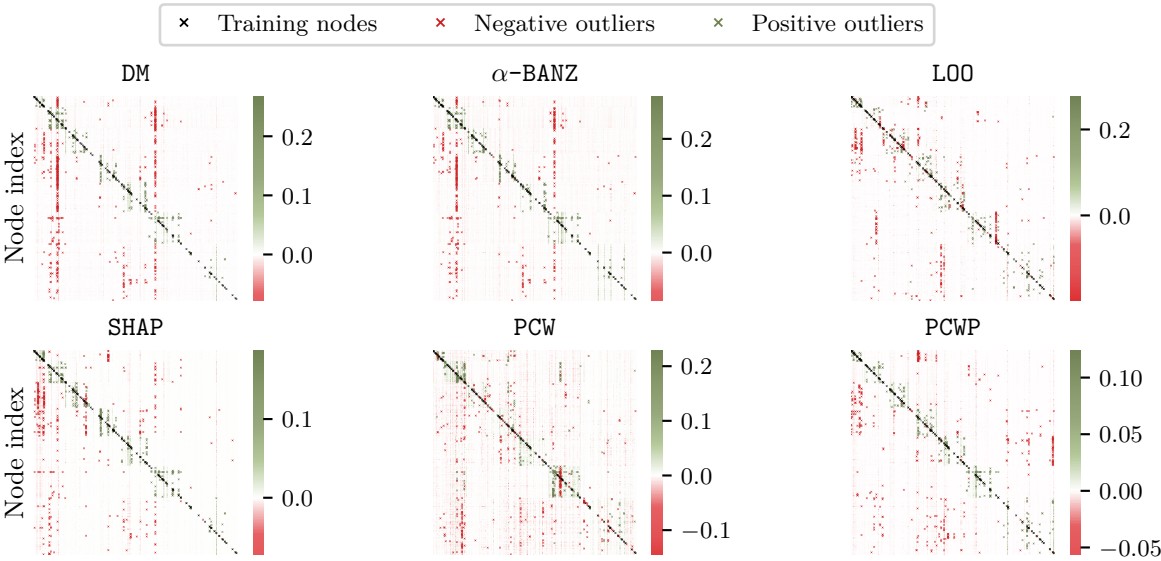

Figure 12: Heatmap of the `learning` signal values $\mathbf{\Phi}$ on `CoraML` for different approaches. Green and red dots represent positive and negative importance respectively. Crosses represent highly impactful nodes (with values larger than the ones indicated in the bars). Black crosses mark the training nodes.

## A   Analysis of node values

Focusing on the values computed with SGC as the model on `CoraML`, we more carefully analyse the computed values in different settings.

### A.1   Node- and class-level influence

**Node influence.**  To provide a more detailed comparison in support of Finding I, we study the values $\mathbf{\Phi}$ under both `learning` and `overall` signals. Fig. 12 and Fig. 13 show the heatmaps of these values, sorted by class size and degree. Positive (green) entries indicate that a node's removal has a beneficial effect on performance, while negative (red) entries show a detrimental effect. Black crosses mark training nodes; their corresponding vertical stripes illustrate how training nodes often exert influence on multiple other nodes supporting also Finding II.

Some methods (e.g., `PCW` and `PCWP`) assign values close to zero under the `overall` signal, mirroring their less reliable performance in the removal experiment. As a result, their final rankings can approximate random noise (see lower plots in Fig. 7).

**Cluster influence.**  Fig. 14 and Fig. 15 show the average $\mathbf{\Phi}$ values on a per-class basis for the `learning` and `overall` signals. We overlay the confusion matrix of the model's predictions, allowing us to interpret how higher positive influence (darker green) correlates with a lower rate of false positives. Notably, `PCW` and `PCWP` have difficulty identifying highly negative influence in the `overall` setting, causing negative values to diminish in magnitude.

Fig. 16 shows the average class influence across all the datasets using $\mathbf{\Phi}_{\alpha\text{-BANZ}}$ (`learning` signal). As pointed out in Finding I, smaller communities get the highest value than larger ones. We see how this effect is consistent across all the considered datasets.

**Node influence visualization.**  Finally, to support Finding III, we extend the breadth-first search influence visualization to the `overall` in Fig. 17. As before, node size encodes importance for a given test node, and the color of the incoming edge (green or red) signifies whether the influence is positive or negative. We again observe that training nodes are typically the most influential, although certain test nodes within the

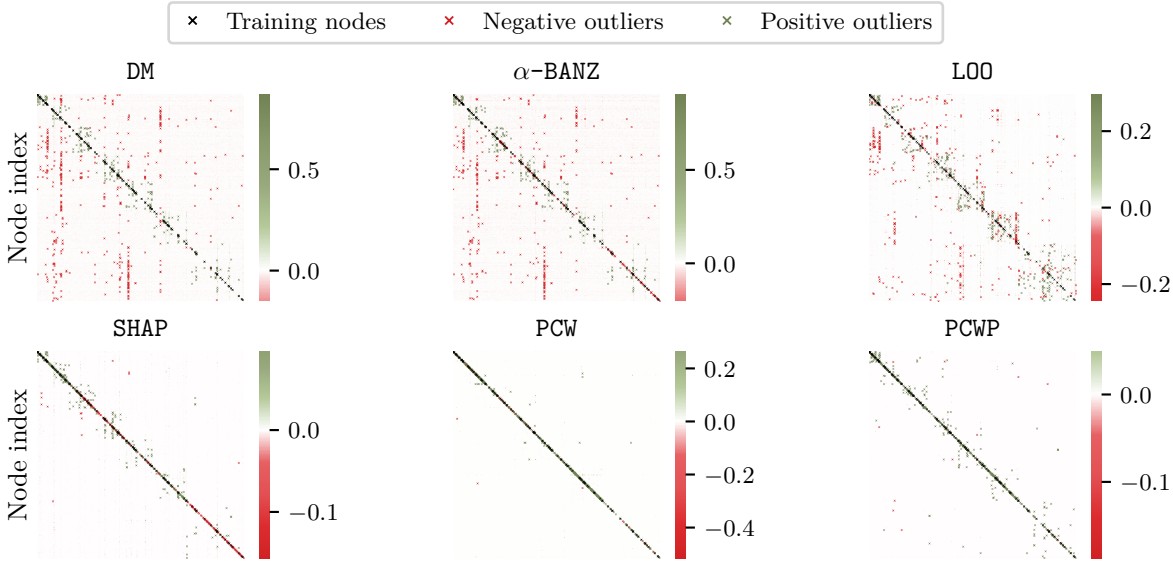

Figure 13: Heatmap of the `overall` signal values $\mathbf{\Phi}$ on `CoraML` for different approaches. Green and red dots represent positive and negative importance respectively. Crosses represent highly impactful nodes (with values larger than the ones indicated in the bars). Black crosses mark the training nodes.

same class (or those tightly connected to highly impactful training nodes) can also assume significant roles. In the `overall` signal, more outliers are apparent in terms of node size, indicating that a few nodes can exert an especially large effect on model outcomes.

### A.2  Synchronized class influence

Analogously to Fig. 5, we investigate how nodes in the same class tend to exhibit similar influence patterns, both in how they influence others and in how they are influenced themselves (Fig. 18). To quantify these effects, we calculate the pairwise $R^2$ coefficient of the rows (bottom) and columns (top) of $\mathbf{\Phi}$ for different experimental settings.

On the left-most column of Fig. 18, we focus on the `train` setting, `overall` signal. As in Finding II, we observe that nodes within class tend to influence other nodes in a similar manner (top), resulting in a pronounced clustering effect. Similarly, the row-wise $R^2$ shows that nodes from the same class are also influenced similarly by other nodes (bottom), aligning with our earlier findings that training nodes can exert a broad impact across the graph.

On the right-most two columns of Fig. 18, we focus on the `all` setting and examine the two different signals. The middle column corresponds to the `learning` signal, while the right column corresponds to the `overall` signal. Similar class-wise clustering patterns are visible in the `learning` plots (particularly for the row-wise $R^2$), but they become weaker under the `overall` signal. We leave this as an interesting direction to investigate more in the future.

Overall, our results highlight a similar influence pattern in how nodes from the same class exert and receive influence to and by others.

### A.3  `Learning` vs. `overall` rank

Because `learning` and `overall` signals can provide complementary insights, we investigate how these two perspectives differ in the rankings they produce. Specifically, for each approach, we compute the overlap in the top k% of node rankings derived from the corresponding $\mathbf{\Phi}$ values for `CoraML`. Fig. 20 summarizes these

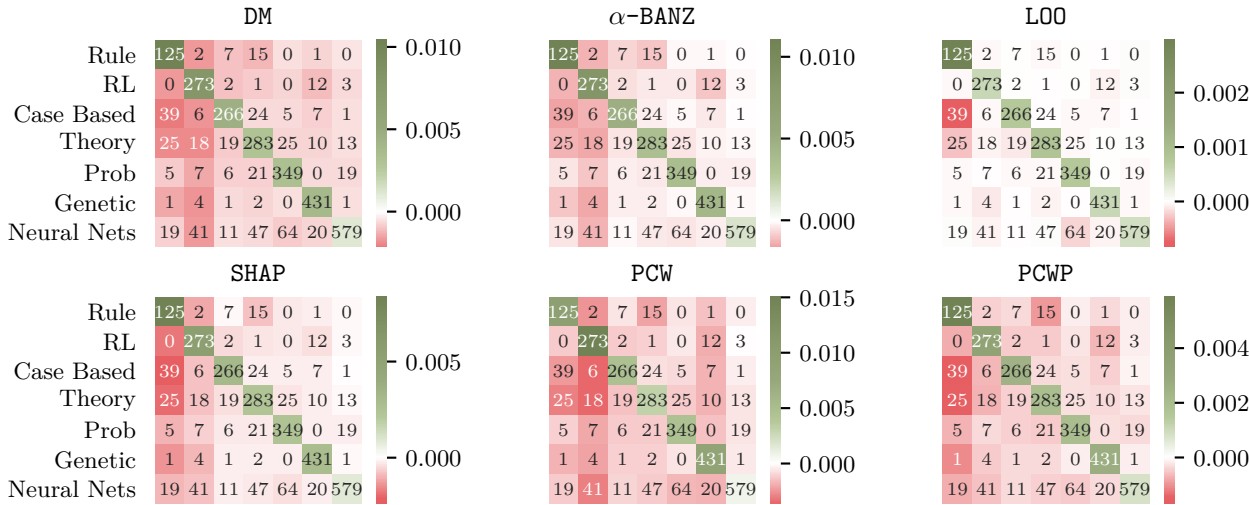

Figure 14: Average class-wise influence across different approaches according to **Φ** capturing the `learning` signal. Classes are sorted by their size. Cell annotations are the confusion matrix values for model predictions.

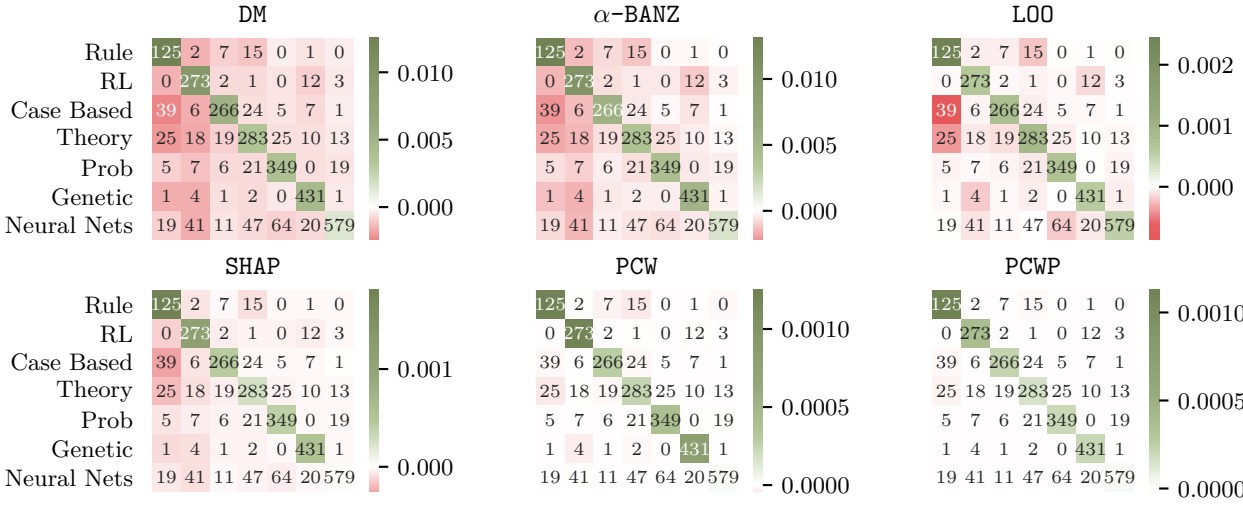

Figure 15: Average class-wise influence across different approaches according to **Φ** capturing the `overall` signal. Classes are sorted by their size. Cell annotations are the confusion matrix values for model predictions.

results, illustrating that `DM` and $\alpha$-`BANZ` yield consistently high overlap between the `learning` and `overall` rankings. This stability suggests that, even when a node is removed for both training and inference, these methods capture node importance in a way that remains robust across varying levels of information.

By contrast, `PCW` and `PCWP` show notable variations in their top k% rankings under different removal conditions. These fluctuations underscore the challenge of maintaining stable value estimates for nodes whose significance may shift when the graph and model context change between training and inference.

## A.4 Rankings of approaches

Because `DM` and $\alpha$-`BANZ` demonstrate similar performance, we also examine how their node-level rankings compare to those generated by other methods. In Fig. 21, we report Kendall's $\tau$ correlations computed from

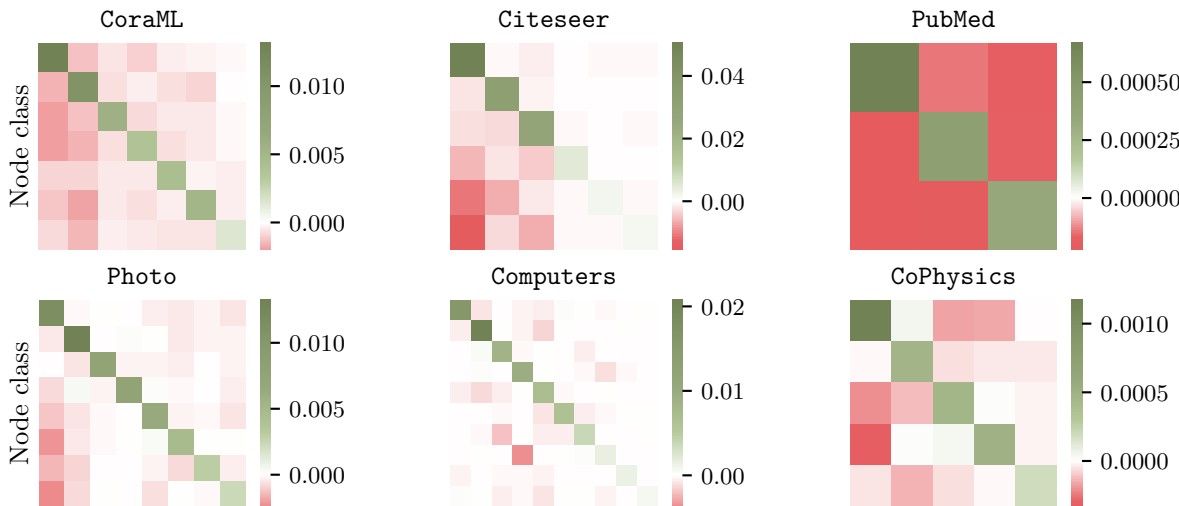

Figure 16: Average class-wise influence from $\mathbf{\Phi}_{\alpha\text{-BANZ}}$ across datasets. Classes are sorted by their size. Adding nodes from smaller communities highly impact the margins positively for nodes from the same class and negatively for nodes from different classes. This effect weakens the larger the community is.

the margin-based $\mathbf{\Phi}$ values for `CoraML`, showing that `DM` and $\alpha$-`BANZ` also exhibit a strong alignment. As expected, `PCW` and `PCWP` show notable correlation too.

Fig. 22 and Fig. 23 illustrate how each method ranks nodes according to the `learning` signal for `CoraML`, using test accuracy and test margins, respectively, as the utility (see § B.1 for an explanation on how to get influence scores from the per-node margins matrix). We highlight training nodes in gray, validation nodes in orange, and test nodes in blue. Although we use the `all` setting (where nodes can be removed at both training and inference), the majority of top-ranked nodes remain in the training set. Nevertheless, certain test nodes appear prominently in these rankings, suggesting that their removal adversely impacts model performance. For instance, `DM` often assigns high importance to particular test nodes which is consistent with Finding III. Overall, these results underscore that while most training nodes drive performance, there can be select test nodes that substantially influence model outcomes as well.

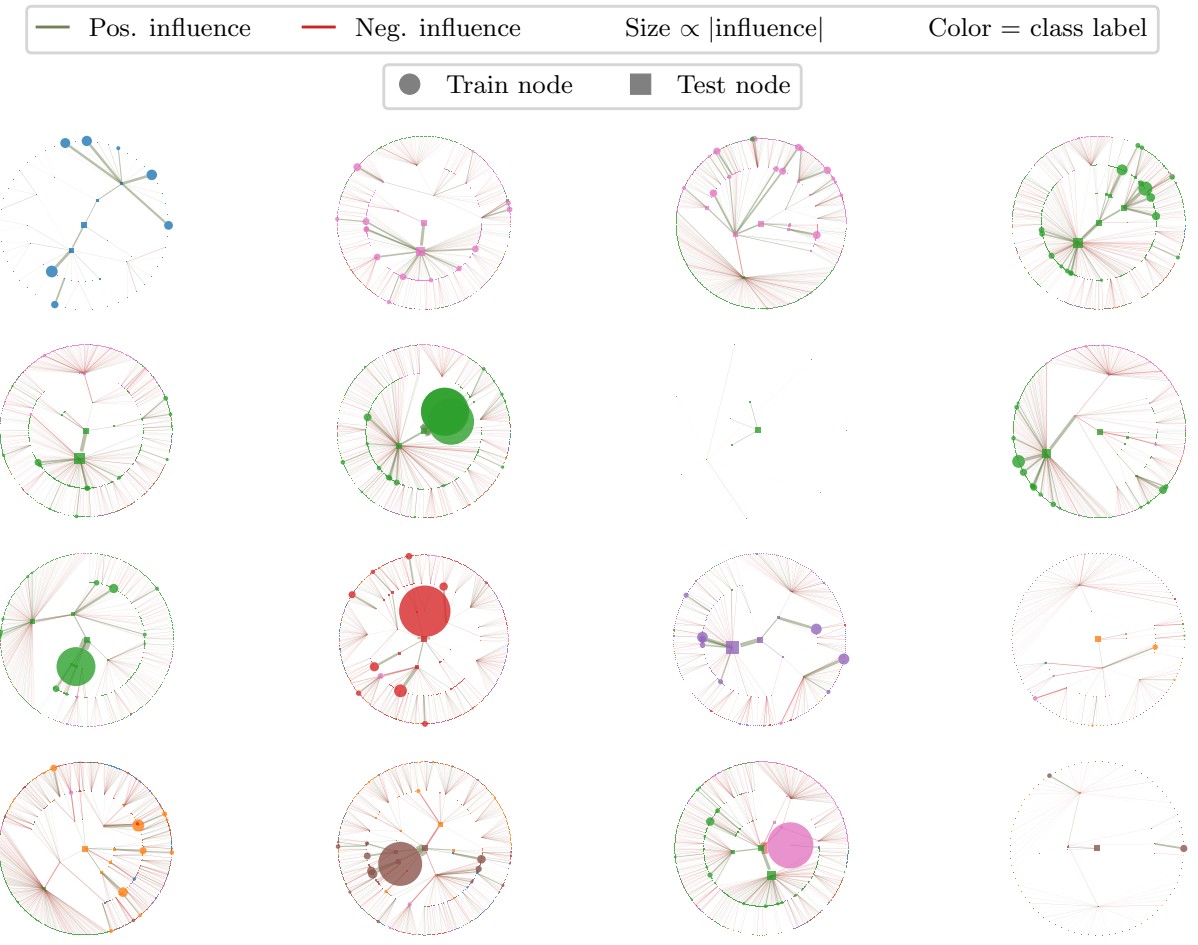

Figure 17: Two-hop breadth-first trees for *overall* signal $\mathbf{\Phi}_{\mathsf{DM}}$. Nodes are class-colored, and edges are colored by sign (green for positive, red for negative). Both nodes and edges are sized based on $\mathbf{\Phi}$ influence values on the root.

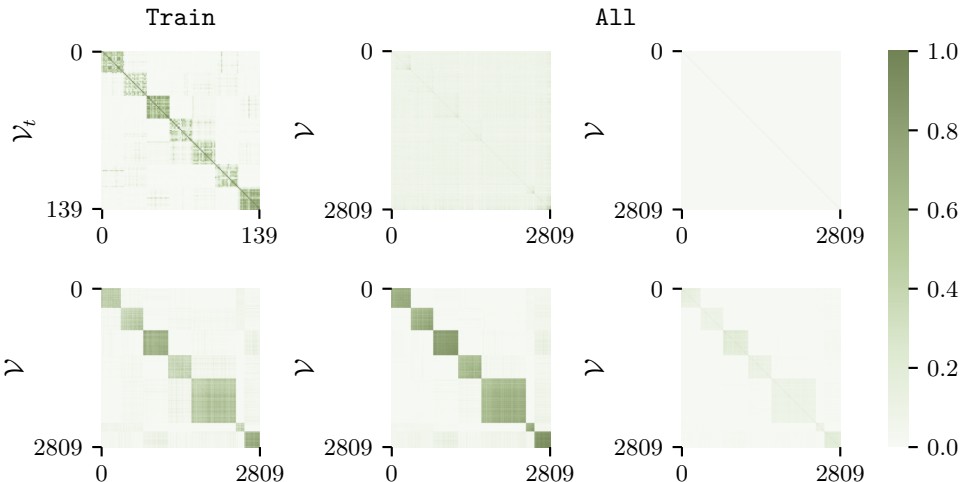

Figure 18: Pairwise $R^2$ coefficients of $\mathbf{\Phi}$ under different settings. The first row shows the column-wise $R^2$ measures, and the second row shows the row-wise $R^2$ measures. The left-most column corresponds to the `Train` setting, `overall` signal. The right-most two columns correspond to the `All` setting, where the middle column represents the `learning` signal and the right column represents the `overall` signal.

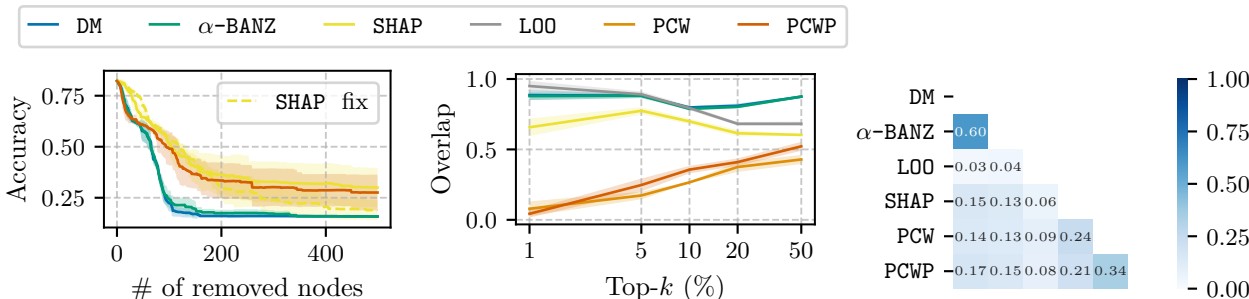

Figure 19: `SHAP` can be improved by adding 10% more nodes as context in its evaluated subsets.

Figure 20: `DM` and $\alpha$-`BANZ` show consistent high overlap between `learning` and `overall` rankings.

Figure 21: Kendall's $\tau$ among approaches indicates `DM` and $\alpha$-`BANZ` share a high rank correlation.

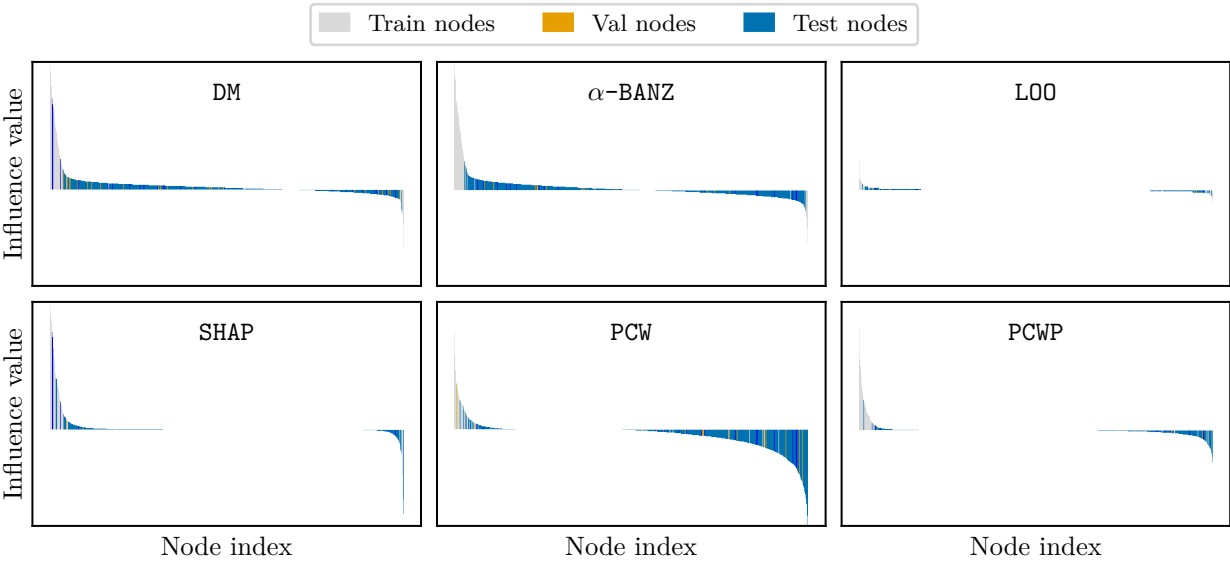

Figure 22: Node rankings from `learning` signal $\mathbf{\Phi}$ for `CoraML` using test accuracy as utility.

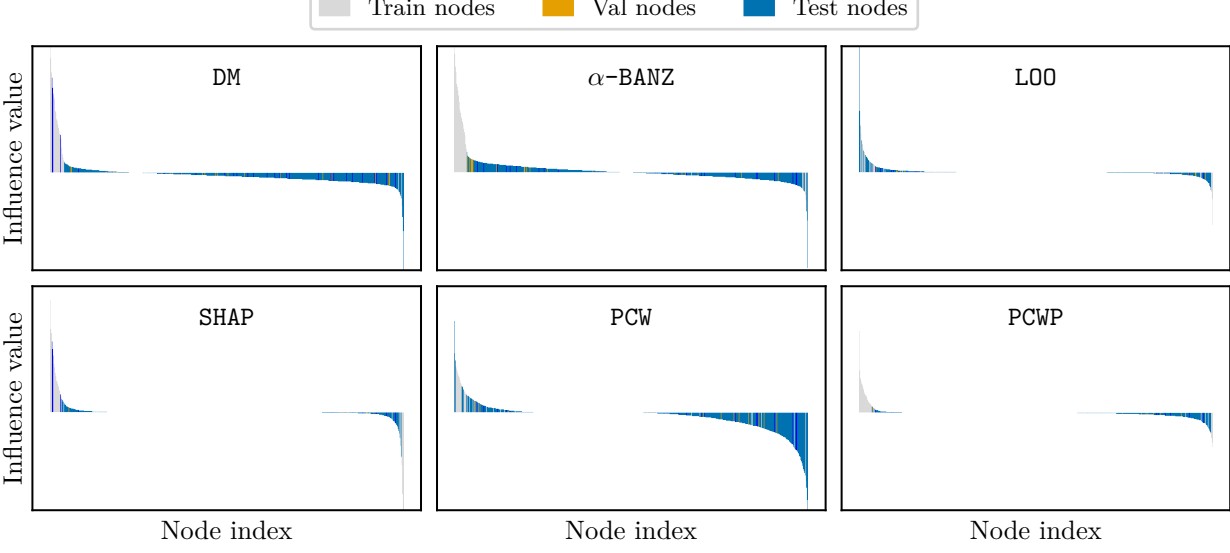

Figure 23: Node rankings from `learning` signal $\mathbf{\Phi}$ for `CoraML` using test margins as utility.

## B  Further experiment details

**Datasets and models.**   Tab. 1 summarizes the datasets used in our experiments. Following Shchur et al. (2018), we employ stratified sampling to select the training nodes (20 per class) along with an equal number of validation nodes. The remaining nodes serve as the test set. We choose SGC, GCN, and GAT as our architectures, each configured with 2 layers. For GCN and GAT, we use 16 hidden channels, a dropout rate of 0.5, and a ReLU activation. Specifically for GAT, we adopt the GATv2 variant from Brody et al. (2022), using 8 attention heads.

Table 1: Summary of datasets. Homophily is computed on the LCC.

| Dataset | Nodes | Edges | Features | Classes | Nodes (LCC) | Edges (LCC) | Homophily | #Train/Val/Test |
|---|---|---|---|---|---|---|---|---|
| CoraML | 2 995 | 16 316 | 2 879 | 7 | 2 810 | 15 962 | 0.784 | 140 / 140 / 2 715 |
| Citeseer | 4 230 | 10 674 | 602 | 6 | 1 681 | 5 804 | 0.928 | 120 / 120 / 3 990 |
| PubMed | 19 717 | 88 648 | 500 | 3 | 19 717 | 88 648 | 0.802 | 60 / 60 / 19 597 |
| Photo | 7 650 | 238 162 | 745 | 8 | 7 487 | 238 086 | 0.827 | 160 / 160 / 7 330 |
| Computers | 13 752 | 491 722 | 767 | 10 | 13 381 | 491 556 | 0.777 | 200 / 200 / 13 352 |
| CoPhysics | 34 493 | 495 924 | 8 415 | 5 | 34 493 | 495 924 | 0.931 | 100 / 100 / 34 293 |
| Texas | 183 | 325 | 1 703 | 5 | 183 | 325 | 0.108 | 87 / 59 / 37 |
| Wisconsin | 251 | 515 | 1 703 | 5 | 251 | 515 | 0.196 | 120 / 80 / 51 |

**Training setting.**   We train GCN and GAT using the Adam optimizer with a learning rate of 0.01 for 3000 epochs. Early stopping is based on the validation loss, with 50 epochs of patience. Results are averaged over 10 runs for each model, except for larger datasets where we use 5 runs. In contrast, SGC leverages a closed-form solution by relaxing the classification objective into a regression problem. Due to the absence of non-linearities, the model weights can be derived as $W^\star = \tilde{X}_\ell H$, where $\tilde{X} = (\hat{X}^\top \hat{X} + \lambda I)^{-1} \hat{X}^\top$ and $\hat{X} = S^2 X$. For all models, we conduct experiments using 5 different train/val/test splits and report average performance. We train each node's datamodel with the ridge regression (with cross-validation) implementation from scikit-learn (Pedregosa et al., 2011) using its default hyperparameters.

Running these experiments on CPUs generally proves faster than on GPUs, given the shallow architectures. We parallelize the runs of different models with the joblib library (Joblib Development Team, 2020), leveraging the larger RAM capacity (compared to GPU memory) to execute as many jobs as possible simultaneously. Experiments are performed on a cluster of 136 nodes, each equipped with 2x AMD Epyc 9654 processors (96 cores, 2.4–3.7 GHz) and 768 GB of RAM. For larger datasets, such as `CoPhysics`, we switch to a bigger partition with 8 nodes (same CPU specifications) but 3 TB of RAM to accommodate higher memory requirements.

**Running times.**   We split the computational cost of each method into two components: gathering utility across the selected subsets and computing the final data values. Tab. 2, Tab. 3, and Tab. 4 show the total cost for each approach under 50 000 subsets with $\alpha = 0.1$ for different GNNs. Among methods with comparable performance, $\alpha$-`BANZ` exhibits the best overall runtime. A "-" in the tables indicates the approach ran out of memory.

Table 2: SGC computation times.

| Dataset | DM | | $\alpha$-BANZ | | LOO | | SHAP | | PCWP | |
|---|---|---|---|---|---|---|---|---|---|---|
| | Train | All | Train | All | Train | All | Train | All | Train | All |
| Citeseer | 7 m 32 s | 11 m 38 s | 7 m 20 s | 5 m 7 s | 13 s | 29 s | 6 m 28 s | 3 m 24 s | 4 m 4 s | 3 m 57 s |
| CoraML | 41 m 30 s | 58 m 50 s | 41 m 13 s | 25 m 57 s | 25 s | 2 m 54 s | 43 m 16 s | 46 m 15 s | 40 m 35 s | 35 m 28 s |
| PubMed | - | - | - | 2 d 4 h 24 m 40 s | 29 m 18 s | 2 d 6 h 57 m 7 s | 2 m 9 s | - | 3 d 17 h 35 m 6 s | 3 d 17 h 42 m 44 s |
| Photo | 7 h 53 m 49 s | 13 h 18 m 7 s | 7 h 46 m 41 s | 3 h 0 m 23 s | 2 m 4 s | 1 h 13 m 57 s | 8 h 25 m 21 s | 17 h 32 m 58 s | 7 h 41 m 21 s | 7 h 42 m 2 s |
| Computers | 1 d 20 h 18 m 23 s | 3 d 16 h 3 m 12 s | 1 d 19 h 39 m 17 s | 16 h 36 m 34 s | 19 m 8 s | 11 h 49 m 18 s | 1 m 8 s | - | 2 d 9 h 30 m 55 s | 2 d 9 h 10 m 48 s |
| CoPhysics | - | - | - | - | 3 h 8 m 27 s | - | 34 m 25 s | - | - | - |

**Datamodels training.**   By default, `DM` excludes any subset that contains a target sample in the training set to prevent information leakage (Ilyas et al., 2022), particularly when evaluating predictive performance. However, this procedure is not always feasible across different graph settings. Specifically, when computing a node's `learning` signal value, we exclude subsets containing that node from the sampled distribution used to

Table 3: GCN computation times.

| Dataset | DM | | α-BANZ | | LOO | | SHAP | | PCWP | |
|---|---|---|---|---|---|---|---|---|---|---|
| | Train | All | Train | All | Train | All | Train | All | Train | All |
| Citeseer | 52 m 40 s | 16 m 33 s | 51 m 6 s | 9 m 46 s | 39 s | 2 m 22 s | 53 m 35 s | 28 m 2 s | 10 m 48 s | 10 m 48 s |
| CoraML | 3 h 53 m 11 s | 56 m 33 s | 3 h 54 m 16 s | 24 m 13 s | 1 m 7 s | 14 m 54 s | 3 h 48 m 52 s | 39 m 35 s | 17 m 39 s | 18 m 1 s |
| PubMed | 3 h 5 m 17 s | - | 3 h 2 m 40 s | 2 h 57 m 21 s | 50 s | 1 h 28 m 26 s | 2 m 4 s | 2 h 20 m 5 s | 43 m 16 s | 44 m 23 s |
| Photo | 10 h 9 m 58 s | 14 h 24 m 27 s | 10 h 17 m 5 s | 34 m 50 s | 9 m 39 s | 5 h 22 m 32 s | 11 h 3 m 58 s | 1 h 44 m 26 s | 21 m 24 s | 21 m 25 s |
| Computers | 1 d 1 h 47 m 17 s | - | 1 d 1 h 20 m 49 s | 1 h 43 m 28 s | 19 m 12 s | 18 h 5 m 51 s | 1 m 8 s | 5 h 22 m 29 s | 27 m 38 s | 27 m 34 s |
| CoPhysics | 4 d 3 h 57 m 31 s | - | - | 1 d 2 h 42 m 20 s | 17 m 23 s | - | 40 m 10 s | - | - | - |

Table 4: GAT computation times.

| Dataset | DM | | α-BANZ | | LOO | | SHAP | | PCWP | |
|---|---|---|---|---|---|---|---|---|---|---|
| | Train | All | Train | All | Train | All | Train | All | Train | All |
| Citeseer | 3 h 40 m 48 s | 25 m 53 s | 3 h 32 m 20 s | 19 m 36 s | 2 m 1 s | 10 m 22 s | 3 h 28 m 29 s | 1 h 0 m 43 s | 27 m 57 s | 27 m 57 s |
| CoraML | 15 h 15 m 20 s | 1 h 17 m 7 s | 15 h 2 m 52 s | 45 m 14 s | 4 m 56 s | 1 h 11 m 2 s | 16 h 50 m 56 s | 1 h 59 m 5 s | 44 m 45 s | 44 m 45 s |
| PubMed | 17 h 59 m 22 s | - | 17 h 46 m 60 s | 3 h 22 m 10 s | 4 m 32 s | 10 h 56 m 16 s | 2 m 12 s | 9 h 58 m 24 s | 52 m 32 s | 47 m 53 s |
| Photo | - | 12 h 30 m 27 s | - | 1 h 17 m 51 s | 44 m 25 s | 1 d 5 h 18 m 20 s | 4 d 7 h 42 m 0 s | 7 h 9 m 39 s | 41 m 51 s | 41 m 33 s |
| Computers | - | - | - | 3 h 50 m 26 s | 2 h 17 m 30 s | - | 1 m 3 s | 1 d 1 h 51 m 20 s | 1 h 14 m 26 s | 1 h 14 m 21 s |
| CoPhysics | - | - | - | 1 d 12 h 25 m 59 s | 1 h 13 m 2 s | - | 38 m 14 s | - | - | - |

train the datamodel.[3] Conversely, when computing the `overall` value, excluding those subsets would leave no data available for training the datamodel, making this approach impractical.

**Banzhaf margins utility.** In our work, we adopt the MSR implementation to approximate the Banzhaf values:

$$\phi_{\texttt{BANZ}}(i) = \frac{1}{|\mathcal{S}_{\in i}|} \sum_{\mathcal{S} \in \mathcal{S}_{\in i}} u(\mathcal{S}) - \frac{1}{|\mathcal{S}_{\notin i}|} \sum_{\mathcal{S} \in \mathcal{S}_{\notin i}} u(\mathcal{S}). \tag{3}$$

While this is straightforward when each node's utility is defined for every other node, when computing the `overall` signal in the `all` setting, the utilities may be undefined for some nodes. Consequently, the sum of utilities has to be averaged according to the computed utilities. Namely, the aggregation depends on whether nodes are inside or outside the subsets considered for node $i$. Concretely, given arrays of margins $\mathbf{\Phi}_{(\mathcal{S})}[i,:]$ for a node $i$ across different subsets $\mathcal{S}$, utilities are undefined for a node $j$ that is not in $\mathcal{S}$. Thus, dividing by $|\mathcal{S}_{\in i}|$ in cases where not all those subsets actually define a margin for $j$, artificially reduces the measured impact of $i$ on $j$. For this reason, we instead average the influence accounting for the the number of times node $j$ appears in the considered subsets $\mathcal{S}_{\in i}$, and analogously for $\mathcal{S}_{\notin i}$.

## B.1 Nodes ranking - from margins to influence scores

In the node influence experiment, each node needs a single influence score. However, when margins serve as the utility, we obtain an influence value for every other node's prediction, $\mathbf{\Phi}[i,:] \in \mathbb{R}^n$. To derive a final score and subsequently a ranking, we average each node's contribution across all other nodes' predictions. Concretely, we form a matrix of influence scores where row $i$ indicates how other nodes influence node $i$'s prediction. We then compute the mean of each column in this matrix to obtain a final score for each node, which we use to produce the ranking.

## B.2 Subsets sampling

Because it is infeasible to evaluate utility for all $2^{|\mathcal{D}|}$ subsets, a common approximation strategy is to sample subsets. A straightforward but effective approach is to include each instance independently with probability $\alpha$, i.e., $\Pr[i \in \mathcal{S}] = \alpha$ for every $i \in \mathcal{D}$. As a result, the subset size follows a Binomial distribution with expected value $\alpha \cdot |\mathcal{D}|$, implicitly capping the average subset size. We adopt this method for both `BANZ` and `DM` values. As shown in § B.2.1, $\alpha$ is an important hyperparameter that we tune. To approximate the `SHAP` values, we randomly sample a permutation of the nodes, then scan them in order until reaching a user-defined truncation threshold. For a fair comparison, each method uses the same total number of subset evaluations.

---

[3]In the `all` setting, this exclusion applies to $\mathcal{V}_u$ as well.

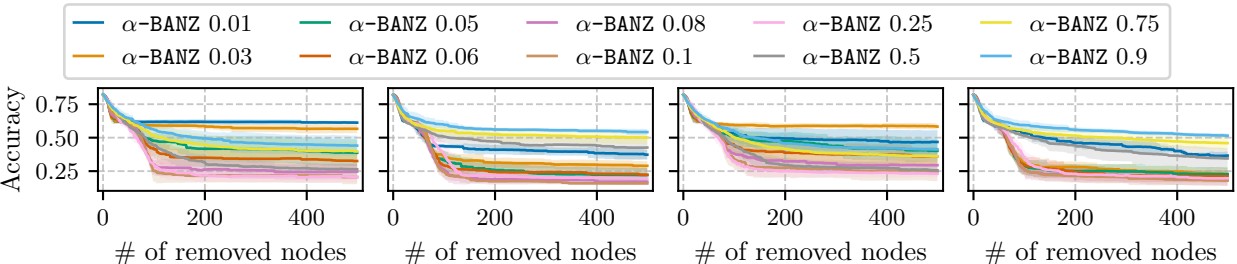

Figure 24: Hyperparameter search for the best $\alpha$ for subsets' sampling on `CoraML` in the `all` setting. From left to right, subplots report the results for `learning` & test margins utility, `learning` & test accuracy utility, `overall` & test margins utility and `overall` test accuracy utility.

We sample 50 000 subsets per method. For $\alpha$-`BANZ` and `DM`, we tested various $\alpha$ values and identified $\alpha = 0.1$ as optimal for `CoraML` (details in § B.2.1); we then apply this choice to the remaining graphs. For `SHAP`, we adopt the implementation of Ghorbani & Zou (2019), which stops processing permutations after 25% of the nodes have been scanned. Consequently, we set the number of permutations to $\frac{50\,000}{\lceil n \cdot \text{tr} \rceil}$ with $\text{tr} = 0.25$. When computing node values under the `all` setting, permutations may start with nodes outside the training set, causing random predictions to introduce noise. Therefore, we begin updating node values only once we reach the first training node in each permutation and truncate 25% from that point. Similarly, for `PCW`, we choose the number of permutations so that the total number of subsets remains 50 000. In Chi et al. (2025a), the default truncation ratios tr for 1-hop and 2-hop neighbors are 0.5 and 0.7, respectively; we optimize these ratios to obtain `PCWP`, which yields slightly better performance (see § B.2.2).

### B.2.1 Choice of the hyperparameter $\alpha$

$\alpha$ is the probability of including each node from $\mathcal{D}$ in a subset, effectively determining the subset size. For example, in the `train` setting, where $\mathcal{D} = \mathcal{V}_t$, choosing $\alpha = 0.1$ yields subsets comprising 10% of $\mathcal{V}_t$ plus all the other nodes. This parameter serves as a hyperparameter for `DM` and $\alpha$-`BANZ` and can be selected via a grid or random search. Due to the high computational cost of performing such a search on every dataset, we determined the optimal $\alpha$ for `CoraML` and reused it across all other datasets, acknowledging it may be suboptimal in certain cases. Nonetheless, as demonstrated in § 4, this choice of $\alpha$ outperforms other methods on most datasets.

Fig. 24 shows how $\alpha$-`BANZ` performs in the most influential pruning experiment under various $\alpha$ values used to sample subsets for node-value prediction. We observe that $\alpha = 0.1$ and $\alpha = 0.25$ yield similar estimates. However, because smaller subsets reduce training time for the models, we chose $\alpha = 0.1$ as the best overall setting.

### B.2.2 Optimized `PCW`

In `PCW`, we compute marginal contributions for nodes only within the first $(1 - \text{tr})$ portion of each node's child sub-trees. We approximate the contributions of the remaining sub-trees as 0, stopping once we reach exactly 50 000 subsets. Because this approach was originally developed for an inductive setting, its default truncation ratios appeared overly tailored to that context. We discovered that optimizing these truncation ratios to maximize the number of considered permutations improved performance in the transductive setting. Consequently, we set both the 1-hop and 2-hop neighbor truncation ratios to 0.99, referring to this variant as `PCWP`.

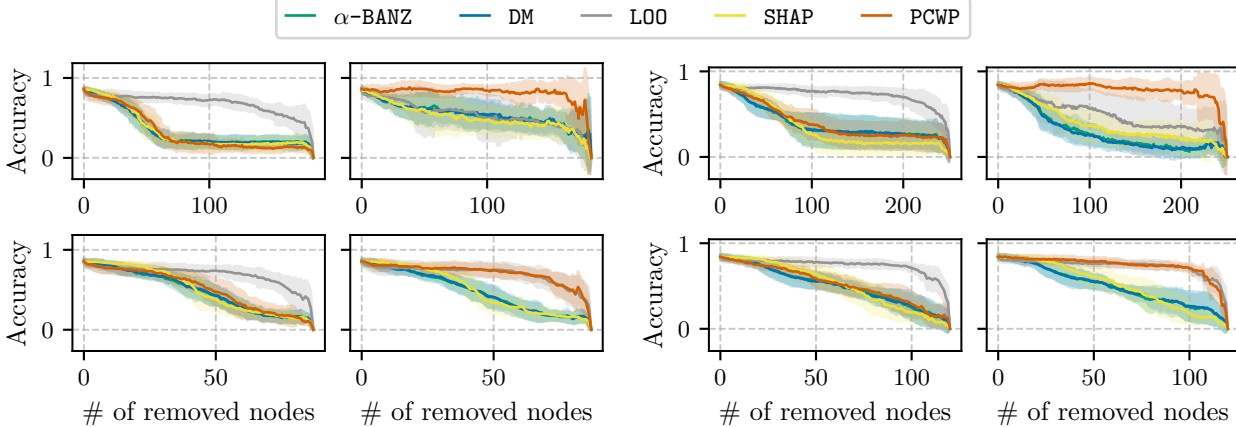

Figure 25: Node drop for `Texas` dataset. Rows correspond to the two valuation settings (`all` at the top, `train` at the bottom); columns to the two utility signals (left: `learning`, right: `overall`). Steeper initial declines indicate more faithful value estimates.

Figure 26: Node drop for `Wisconsin` dataset. Rows correspond to the two valuation settings (`all` at the top, `train` at the bottom); columns to the two utility signals (left: `learning`, right: `overall`). Steeper initial declines indicate more faithful value estimates.

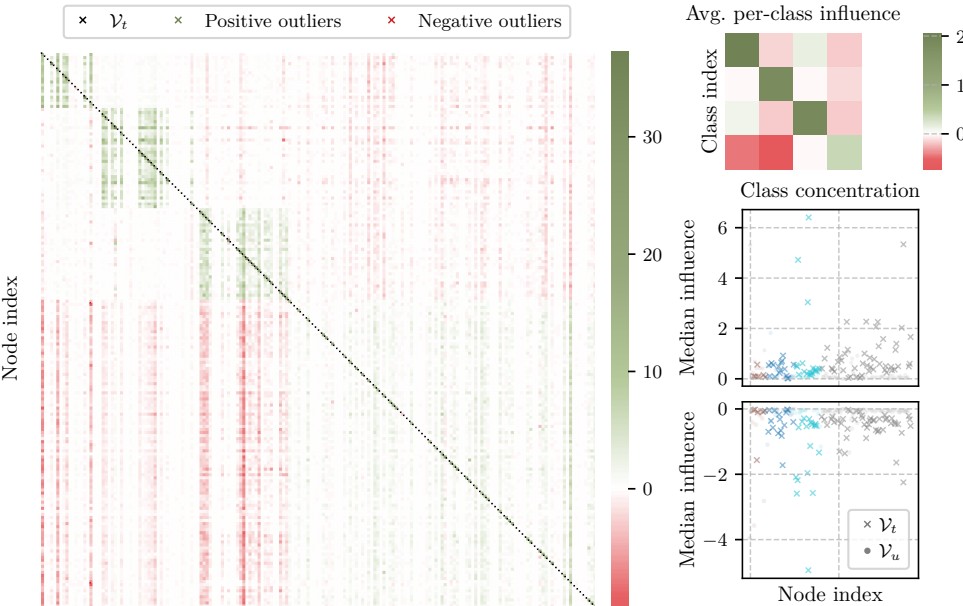

Figure 27: $\Phi_{\text{DM}}$'s heatmap on `Texas` graph (`all-learning` scenario). Rows and columns are sorted by class, cluster size and degree. Green (red) values mean that adding column-node $j$ increases (decreases) the margin of row-node $i$. We can see positive intra-class influence, but weaker mixed off-diagonal colors, confirmed also by the average per-class influence heatmap. Influence is concentrated in a few (training) nodes.

## C    Additional experiments

### C.1    Heterophilic datasets

To check whether our findings still hold in the heterophilic settings, we repeated our analysis on the `Texas` and `Wisconsin` (Pei et al., 2020) hyperlink-web graphs, where the homophily ratio is less than 20% (see

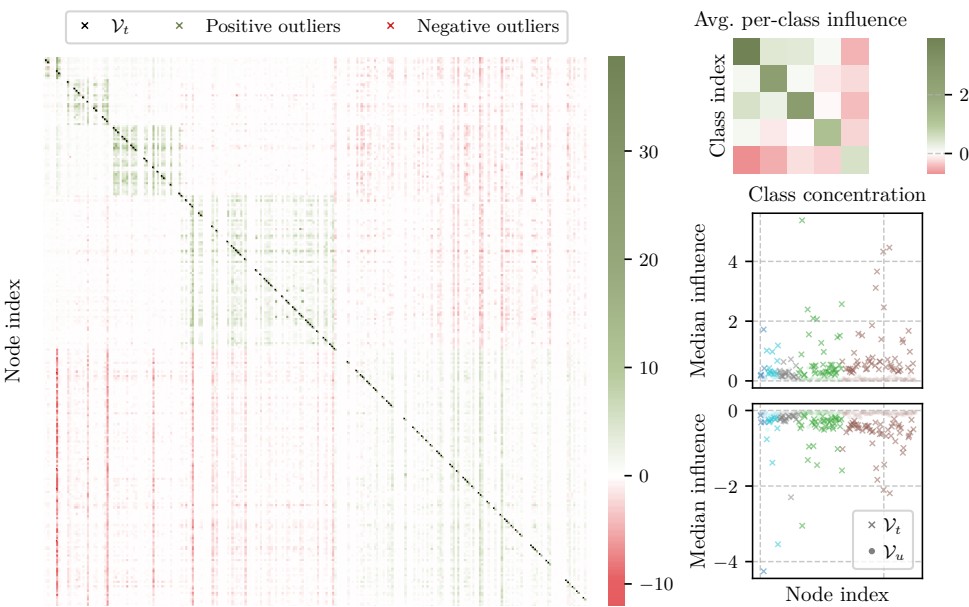

Figure 28: $\mathbf{\Phi}_{\text{DM}}$'s heatmap on `Wisconsin` graph (`all-learning` scenario). Rows and columns are sorted by class, cluster size and degree. Green (red) values mean that adding column-node $j$ increases (decreases) the margin of row-node $i$. We can see positive intra-class influence, but weaker mixed off-diagonal colors, confirmed also by the average per-class influence heatmap. Influence is concentrated in a few (training) nodes.

Tab. 1). The experimental protocol, hyper-parameters and number of sampled subsets ($50\,000$, with $\alpha = 0.1$ for sampling-reuse methods) are exactly the same as in the homophilic benchmarks. We used the 10 different node splits that are provided with the datasets. Finally, we figure out that in the `Texas` dataset, there is only one node belonging to class 1. We then omit this class when reporting the per-class average influence.

Fig. 25 and Fig. 26 show the loss in accuracy we observe when cumulatively removing the top-valued nodes. $\alpha$-BANZ and DM again show the largest accuracy drop across different settings (`all` setting on top and `train` setting at bottom vs `learning` signal on the left and `overall` signal on the right). The shape of the drop curves mirrors what we saw on homophilic graphs: an initial steep fall followed by a plateau once low-value nodes start to dominate. The plateau is reached sooner here because both datasets are an order of magnitude smaller.

In Fig. 27 and Fig. 28, we repeat the same analysis performed in Fig. 4. We show heatmaps of the values $\mathbf{\Phi}_{\text{DM}}$ in the `all` setting using the `learning` signal, and still observe clear positive blocks along the diagonal (nodes help peers of the same class), but the off-diagonal entries are no longer strongly negative. In some small classes they even turn slightly positive, reflecting the fact that cross-class neighbors are the norm rather than the exception on these graphs. Consistent with the homophilic case, influence is highly

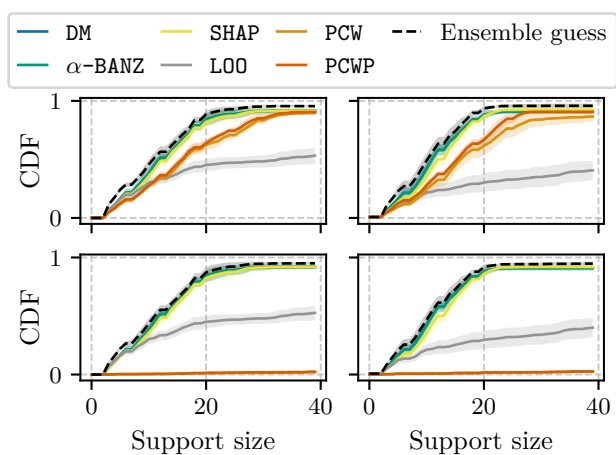

Figure 29: Brittle prediction in the `train` setting for SGC (left) and GCN (right). DM and $\alpha$-BANZ consistently approach the best guess regardless of the considered `learning` (top) or `overall` (bottom) signals.

concentrated: in both datasets fewer nodes have the most of the influence value, and most of these are still training nodes.

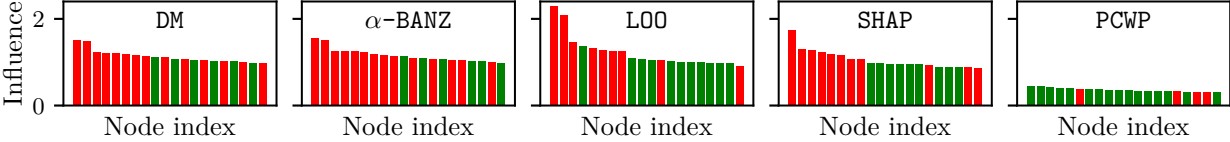

Figure 30: Nodes rankings according to values computed on a graph with 10% of poisoned (red) nodes.

### C.2 Brittleness in the `train` setting

We replicate the same experiment used in Finding IV, but now in the `train` setting, to assess whether each approach maintains strong predictive performance. In Fig. 29, we evaluated both the `learning` (top) and `overall` (bottom) signals computed with SGC (left) and GCN (right), and in each scenario, DM and $\alpha$-BANZ demonstrated performance closest to the best guess among all methods.

### C.3 Memorization ranking

Fig. 30 shows the rankings of the approaches as a result of the memorization experiment. As explained in Finding IV, we poison 10% of the training data and expect a good data validation approach to rank these nodes as the most important for their prediction (self-importance). We show the results for `CoraML`, where the number of training nodes is 140 resulting in 14 poisoned nodes. We then show only the top 14 nodes ranked by the approaches. The results show that DM and $\alpha$-BANZ rank first the poisoned nodes establishing as robust approaches for detecting poisoned or mislabeled data.

### C.4 Influential nodes in the `train` setting

Here we present the results for the node influence experiment in the `train` setting, namely we attribute importance (and then remove) only training nodes. In Fig. 31, we show how the approaches capture the real value considering both the `learning` (top) and `overall` (bottom) signals, across different models – SGC (first column), GCN (second column) and GAT (third column). Notably, we observe the approaches struggle more in attributing the right values when non-linearities are introduced in the models (compare SGC w.r.t. GCN and GAT). Consistently with previous results, DM and $\alpha$-BANZ perform better than the other approaches.

In this experiment, we train DM by considering all possible node subsets, regardless of whether a particular node is included. Unlike the procedure described in § B, this setup is feasible here because we are no longer focusing on maximizing predictive performance (where information leakage would pose a risk). Instead, our primary goal is to evaluate each method's ability to identify problematic data.

### C.5 Most influential addition

In Fig. 32, we show the results for the most influential node addition experiment. In this experiment, we start with an empty graph and then incrementally add nodes in order of their importance, as determined by each method's computed node values $\mathbf{\Phi}$. We evaluate the predictive performance (test accuracy) of the model after each batch of added nodes. Contrary to the most influential pruning, because we are adding most influential nodes first, we anticipate a rapid initial increase in accuracy, followed by a plateau once most of the crucial nodes have been included. We present results computed according to both `learning` (top) and `overall` (bottom) values. Similarly to the removal experiment, DM and $\alpha$-BANZ better assess node values.

### C.6 Stability

A crucial aspect of reliable data valuation is consistency – producing similar ranking results across multiple runs of the same experiment. Fig. 33 illustrates the stability of each approach, measured over 10 random seeds for each of the 5 train/val/test splits in `CoraML`. Each seed defines different subsets/permutations of nodes

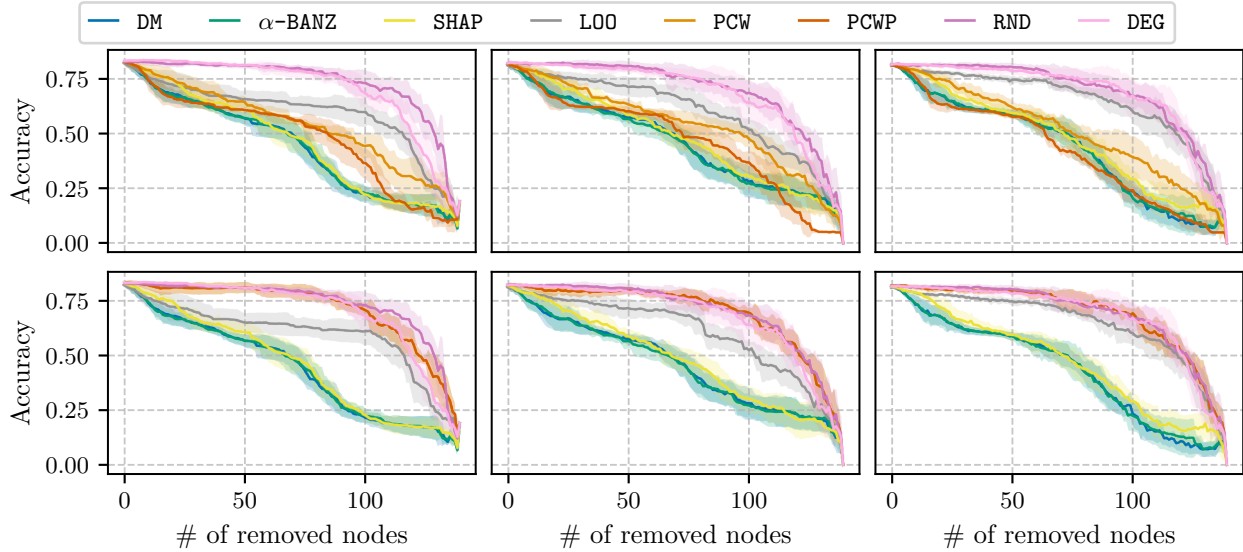

Figure 31: Most influential node removal for `CoraML` across models in the `train` setting: SGC (first column), GCN (second column) and GAT (third column). `learning` signal (top) and `overall` signal (bottom) $\Phi$.

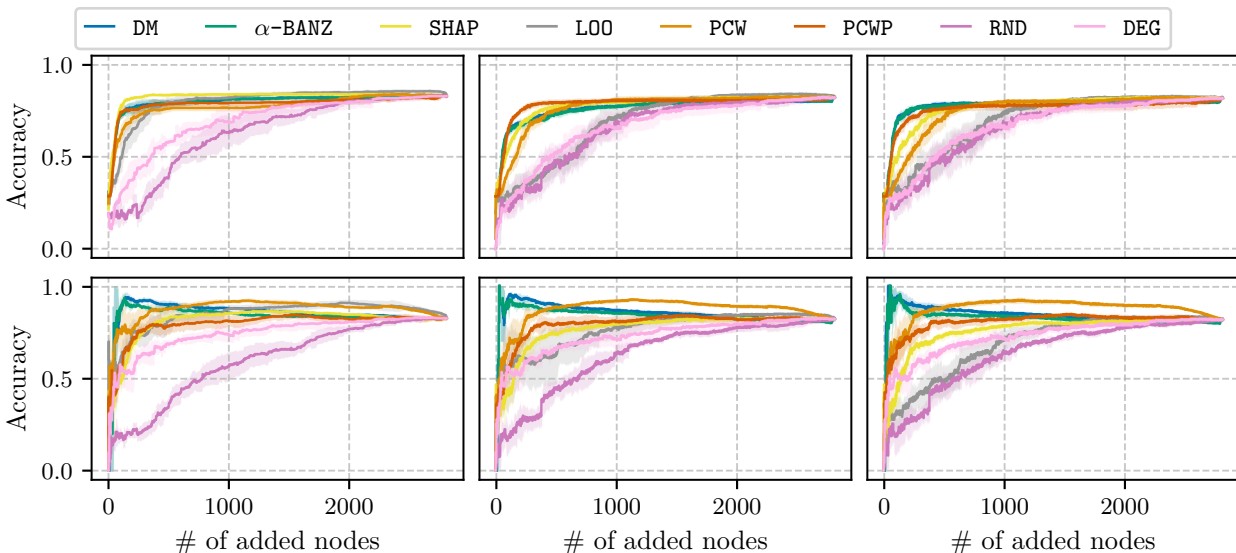

Figure 32: Most influential node addition for `CoraML` across models: SGC (first column), GCN (second column) and GAT (third column). `learning` (top) and `overall` (bottom) signal of $\Phi$. Adding high-value nodes leads to a performance (test accuracy) raise until reaching a plateau given by keep adding uninfluential nodes.

for valuation. We report the average performance of each method (left) and observe that `DM` and $\alpha$-`BANZ` exhibit the lowest variance across runs, making them the most reliable choices for data valuation on graphs.

We also evaluate ranking stability (right panel) by measuring the percentage of top-k nodes that overlap across different seeds in `CoraML`. Once again, `DM` and `BANZ` demonstrate near-identical rankings across all seeds for a given train/val/test split, underscoring their reliability and consistency.

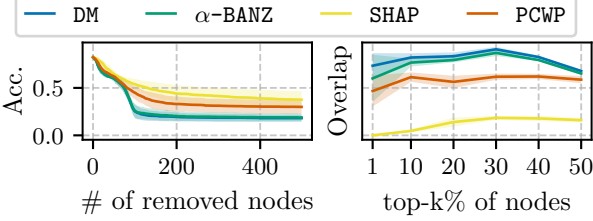
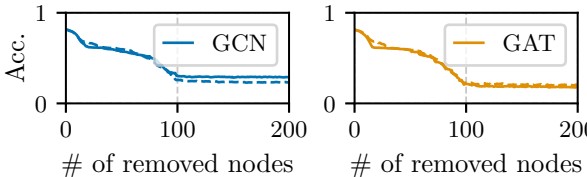

Figure 33: Approaches stability: (left) in predicting node values; (right) in predicting the ranking for a given split across 10 seeds.

Figure 34: Node removal using data values computed and evaluated by each model (solid line) or transferred from SGC (dashed line).

## C.7 Transferability

In Fig. 34, we repeat the node removal experiment (similar to Fig. 8) with two variants: i) Computing data values with a GCN model and evaluating performance on the same GCN (solid line, no transfer). ii) Computing data values with SGC and seeing whether these values transfer (dashed line) to GCN and GAT.

Using $\alpha$-BANZ and margins as the utility metric on CoraML, we observe nearly identical performance among these variants, indicating that data values capture intrinsic properties of the data rather than the specifics of each model. This finding further suggests that cheaper models (e.g., SGC) suffice to compute data values that effectively transfer to more complex architectures.

## C.8 Empirical limitation of game-theoretic approaches

We investigated why game-theoretic methods underperform compared to MSR-based approaches. Our analysis suggests that those methods commonly evaluate subsets containing only a small number of nodes when building up from empty sets via permutations. This yields many degenerate subgraphs and introduces noise in the final node value estimates. In contrast, randomly sampling larger subsets seems to better capture the underlying graph structure.

To address this, we propose a small modification to SHAP that ensures every evaluated subset includes a bit more context. Specifically, whenever a node is added to the subset during the permutation scan, we also include an extra 10% of the total nodes (in line with $\alpha = 0.1$ in MSR-based approaches). As in the standard data Shapley procedure, only the value of the newly added node is updated; the difference is that the node's influence is now assessed in a richer structural context.

Fig. 19 illustrates the impact of this modification: the dashed line shows that even a modest increase in contextual nodes yields more realistic Shapley values, thereby boosting performance. Determining the optimal fraction of additional context nodes remains an open question and is left as future work.

## C.9 Unstructured data values

We assess the contribution of graph connectivity by comparing a standard two-layer GCN with an "unstructured" variant whose adjacency matrix is replaced by the identity, so that every node only sees itself. We keep the same optimizer, initialization, and train/val/test splits for both the two models. Because the identity graph leaves each node isolated, any difference we observe must come from information that the CoraML graph propagates into the node representations.

After training, we compute the learning influence values $\boldsymbol{\Phi}$ (real graph) and $\boldsymbol{\Phi}_{\mathrm{id}}$ (identity) in the all setting, take the median influence value that a node $v$ exerts on others, and plot the difference $\Delta(v) = \mathrm{median}\big(\boldsymbol{\Phi}[v,:]\big) - \mathrm{median}\big(\boldsymbol{\Phi}_{\mathrm{id}}[v,:]\big)$ in Fig. 35. The left panel shows that message passing amplifies positive influence, especially for training nodes, whereas most non-training nodes remain close to zero. The right panel indicates that already negative influences become slightly less negative, with only a few outliers moving in

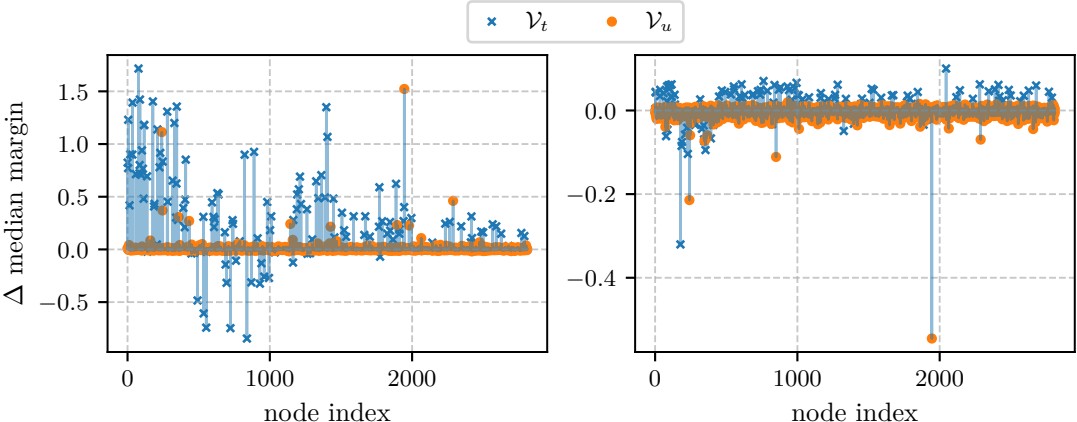

Figure 35: $\Delta$ median influence (GCN – GCN$_{\mathrm{id}}$) for positive (left) and negative (right) `learning` influence values in the `all` setting shows that real connectivity of `CoraML` mainly amplifies training node influence, leaving most non-training nodes unchanged.

the opposite direction. Together these observations confirm that the specific connectivity pattern of `CoraML` amplifies node influence.

