# OpenReview forum: "Node-Level Data Valuation on Graphs"
_TMLR — Accepted by TMLR_

### Review · Reviewer_dCNo · 2025-06-02

**Summary Of Contributions:**

This paper analyzes and evaluates existing data valuation methods on graph neural networks at the node level. It summarizes the experimental results and compares the performance of different valuation methods in a semi-supervised setting.

**Audience:**

Yes

**Broader Impact Concerns:**

## Broader Impact Concerns
This paper compares different data valuation methods for semi-supervised graph neural networks. I did not see any concerns regarding its broader impact.

**Claims And Evidence:**

Yes

**Requested Changes:**

1. Please clearly state the settings for each experiment in Section 4. For instance, I am a bit confused about what model/data value are used in Finding I.

2. The manuscript's overall structure is clear, but the writing could be improved for better clarity.
   1. Several long sentences should be broken up with commas to improve grammatical correctness and enhance readability.
   2. Abbreviations like "SGC" should be defined upon first use.
   3. Minor typos, such as "alos" in the first paragraph of page 5, should be corrected.

**Strengths And Weaknesses:**

### Strengths
1. Data valuation in graph neural networks is an interesting and under-explored area.
2. This paper presents several experiments across several downstream tasks.

---

### Weaknesses
1. The contribution of this work is limited. It can be acknowledged that the authors do not aim to propose novel techniques and instead focus on analyzing existing data valuation methods for GNNs. However, the study neither proposes an objective and measurable standard for benchmarking these methods nor provides actionable insights that could serve as concrete guidelines for future research. Therefore, despite some interesting empirical observations, the contribution is insufficient for this journal.

2. Some conclusions in the Discussion section do not appear well-supported by the current experiments. For example, in the first paragraph, the authors claim that "connectivity patterns amplify directional influence" seems to rely solely on the observed asymmetric influence between classes. However, this asymmetry is both expected and intuitive and is commonly observed even in standard i.i.d. settings. Without additional evidence, it is difficult to conclude that connectivity patterns are the underlying cause of this phenomenon. In addition, the discussion on poisoning in the second paragraph—specifically the statement "This redundancy suggests opportunities for potential pruning, but also vulnerabilities"—is somewhat confusing. If nodes behave as collective units, one would expect the effect of a single poisoned node to be diluted due to redundancy. Therefore, the claim that redundancy introduces vulnerabilities requires further evidence to be convincing.


3. The experimental settings need further clarification. For instance, in Figure 8, how are the 500 nodes selected for removal? According to Table 1, all datasets contain fewer than 200 training nodes. Does this imply that test nodes are also removed? If so, it would be problematic to compare test accuracy across different runs, as the test set is no longer consistent.


4. The authors claim that the previous papers consider only training nodes in a transductive setting, ignoring in such a way the interactions between unlabeled and labeled nodes" (Page 12), which is inaccurate. For example, Chen et al. explicitly consider labeled and unlabeled data, as illustrated in Figure 1 of their paper. The figure shows that unlabeled nodes, represented as smaller nodes, are included in the influence estimation process through edge removal, capturing their interactions with other nodes in the graph.

---

> ### Author Response · Authors · 2025-06-17
>
> We thank the reviewer for finding the investigated topics of interest and for finding our experiments extensive across several downstream tasks. In the following we are going to address the raised concerns.
>
> **(W1) Limited contribution**
>
> As also mentioned in the introduction, our goal is a *systematic benchmark* rather than a new valuation rule. We respect the point of view of the reviewer, but we disagree with saying that we don’t provide actionable insights. Indeed, we have a Discussion paragraph at the end of section 4 where, in addition to qualitative insights, we provide actionable insights like the adoption of MSR-based notions hinted by the better performance across different downstream tasks of such approaches (i.e. datamodel and data banzhaf). In addition, we would like to emphasize the criteria of the TMLR journal based on the technical soundness and clarity of the arguments presented, as well as if findings in the paper are of interest to TMLR’s audience. Based on the received comments, we think the manuscript matches these criteria.
>
> **(W2) Unsupported statements**
>
> We agree with the reviewer that more evidence is needed for stating that the connectivity amplifies directional influence, and for this reason we have removed the statement as it needs further investigation. We would appreciate if the reviewer can share the related works (in the i.i.d. setting) where “asymmetry” was shown in order to appropriately cite them.
>
> To assess more rigorously the role of the structure in the computation of node values, we perform a new experiment where we train a vanilla two-layer GCN and an “unstructured” version where we simply replace the adjacency matrix of CoraML with the identity such that the structure is not involved anymore. We observe that the final influence is amplified when the structure is taken into account by the model. We included this result in Appendix C.9.
>
> We also rewrote the paragraph on *redundancy* to be more accurate. The changed text is marked in blue.
>
> **(W3/RQ1) Clarify experiment settings**
>
> As explained in Finding V, we select the 500 nodes according to their values, i.e. we rank them from the highest value to the lowest and then we train models via cumulatively remove them. When considering the `all` setting (where all nodes are considered for computing the node values, see Section 3) also test/val nodes can be removed in addition to training nodes. For the evaluation, we can either put back all the removed nodes (`learning`) or keep the same subgraph (`overall`). We agree with the reviewer that in the latter scenario, removing test nodes would be problematic for comparing the test accuracy across different runs. However, note that this happens only when computing the `overall` signal, that it is not the case for Figure 8. We include in the manuscript results for the `overall` setting as well for completeness.
>
> **(W4) Prior work**
>
> What was intended there is that the authors never explicitly study what happens when removing potential highly influential test nodes. Indeed they model the influence of edge removal problem as an “instance reweighing problem” where only training instances are involved. We modified accordingly the manuscript to make the sentence more accurate.
>
> **(RQ2) Writing and typos**
>
> We have updated the paper with the requested changes.

---

### Review · Reviewer_BmWp · 2025-06-03

**Summary Of Contributions:**

The authors present an extensive study of data valuation methods for graph-structured data. Comprehensive experiments are conducted on a wide range of datasets, and several interesting findings are summarized. The authors also demonstrate the benefits and potential applications of computing node values, such as identifying influential nodes, quantifying model brittleness, detecting poisoned data, and others.

**Audience:**

Yes

**Broader Impact Concerns:**

None.

**Claims And Evidence:**

Yes

**Requested Changes:**

1. What is the homophily ratio of the used datasets? Datasets with low homophily ratios, like Texas and Wisconsin [1], should also be considered to enhance the claim, especially Key Findings 1.

2. Dataset information is missing in most Figures' captions and the corresponding textual content, making it hard for readers to follow the experimental settings.

3. I notice that the Appendix includes a wide range of experiments, including analyses of detecting influential nodes or abnormal nodes. I suggest that authors should add corresponding notes in the main paper to facilitate the readers, especially considering the considerable
length of this paper.

[1] GBK-GNN: Gated Bi-Kernel Graph Neural Networks for Modeling Both Homophily and Heterophily. WWW 2022.

**Strengths And Weaknesses:**

**Strength**:

1. Overall, this paper is well written and easy to follow.

2. Node valuation is an interesting topic and is significant to the community.

3. I admire the comprehensive experiments conducted by the authors, which I believe can well support the key claims in this paper.

**Weakness**:

1. What is the homophily ratio of the used datasets? Datasets with low homophily ratios, like Texas and Wisconsin [1], should also be considered to enhance the claim, especially Key Findings 1.

2. Dataset information is missing in most Figures' captions and the corresponding textual content, making it hard for readers to follow the experimental settings.

3. I notice that the Appendix includes a wide range of experiments, including analyses of detecting influential nodes or abnormal nodes. I suggest that authors should add corresponding notes in the main paper to facilitate the readers, especially considering the considerable
length of this paper.

4. I agree that the method is computationally intensive, as the authors claim in Section 6, which I believe is a bottleneck for further application, especially for large graphs with millions of nodes.

[1] GBK-GNN: Gated Bi-Kernel Graph Neural Networks for Modeling Both Homophily and Heterophily. WWW 2022.

---

> ### Author Response · Authors · 2025-06-17
>
> We thank the reviewer for the positive assessment of our writing, topic relevance, and experimental breadth. Your concerns are addressed below.
>
> **(W1/RQ1) Add low homophily graphs**
>
> Table 1 now lists the homophily ratio of every graph. Following your suggestion, we added the heterophilic Texas and Wisconsin WebKB graphs. Results are in Appendix C.1 “Heterophilic datasets”. Finding I still holds: namely, for heterophilic datasets as well, within-class influences are predominately positive. However, the across-class *negative* influence is weaker (occasionally even slightly positive) on heterophilic graphs. We perform the node drop experiment too and DM and α-Banzhaf remain the best-performing valuation across settings.
>
> **(W2/RQ2) Missing dataset and model info**
>
> We agree with the reviewer that dataset and model information is sometimes missing in both text and captions. We worked to improve this in the new updated manuscript and ensured that all the figures have their setting explained at whole. Please let us know if we missed to mention the setting in some figure.
>
> **(W3/RQ3) Reference extensive appendix**
>
> We thank the reviewer for the suggestion and we updated the manuscript to appropriately reference appendix in the text with a brief explanation of what will be found in the referenced subsection.

---

### Review · Reviewer_GAFF · 2025-06-09

**Summary Of Contributions:**

This work provides an empirical study on existing data valuation techniques. The study constructs various scenarios to understand how both labeled and unlabeled nodes influence GNN training and inference. The resulting node values effectively identify influential nodes, quantify model brittleness, detect poisoned data, and predict counterfactuals. The results further highlight that data Banzhaf and datamodels outperform graph-specific methods.

**Audience:**

Yes

**Claims And Evidence:**

Yes

**Requested Changes:**

- As pointed out by the weaknesses, maybe add (1) some discussions on the limitations of the evaluation strategies and (2) some insights about dataset construction and algorithm design.

**Strengths And Weaknesses:**

**Strengths**

- This work provides a comprehensive study of data valuation for graph-based models in semi-supervised settings, rigorously evaluating existing methods not previously applied to graphs.
- The paper evaluates a broad range of methods, including game-theoretic and predictive (DM) approaches, alongside graph-specific methods.
- The analysis yields several key findings on node influence dynamics, covering detection of brittle/poisoned nodes and the importance of sample reuse in graph data valuation.

**Weaknesses**

- The current approach to node removal for data valuation makes it challenging to disentangle the distinct influences of node features versus graph structure (edges), especially concerning noise sources.
- While providing valuable insights, the paper could offer more explicit guidance on how these observations can be leveraged to design improved graph datasets or more robust and efficient GNN algorithms.

---

> ### Author Response · Authors · 2025-06-17
>
> We thank the reviewer for finding our evaluation setting rigorous. In the following, we addressed the raised concerns.
>
> **(W1) Feature vs. structure influence**
>
> We agree that our current evaluation entangles node features and incident edges (see the Setting paragraph of Section 4): when a node is omitted, both signals vanish, so we only report the total value $\phi(v) = \phi^{\mathrm{feat}}(v) + \phi^{\mathrm{adj}}(v)$  as a simpler starting point. Disentangling these two components would require a different experimental protocol (e.g., feature masking vs. edge removal), which we view as important future work. We have added an explicit note to Section 4 clarifying more this limitation.
>
> **(W2) Dataset construction and algorithm design**
>
> In the Discussion paragraph at the end of Section 4, we list a few concrete actions based on the insights from our analysis. We might for example i) prune nodes with strongly negative learning-signal values to sanitise training data without harming accuracy, or ii) manually inspect high self-valued nodes to spot poisoned data. However, as our study is data-centric we can’t see direct ways for improving algorithms hinted by the insights.

---

### Review · Reviewer_ZMfm · 2025-06-09

**Summary Of Contributions:**

This paper presents the first comprehensive empirical study of data valuation methods for graph-structured data in semi-supervised transductive settings. The authors systematically evaluate existing data valuation approaches including game-theoretic methods like data Shapley and data banzhaf, predictive methods like datamodels, and the recently proposed graph-specific precedence-constrained winter value on node classification tasks. Key contributions include (1) adaptation of i.i.d. data valuation methods to graph settings with careful consideration of labeled/unlabeled node interactions, (2) introduction of learning vs. overall signal distinction for graph data valuation, (3) extensive empirical evaluation across multiple datasets and models, and (4) five key findings about node influence patterns, brittleness detection, and the importance of sample reuse in graph data valuation.

**Audience:**

Yes

**Broader Impact Concerns:**

The work focuses on methodological evaluation of data valuation techniques and does not raise significant ethical concerns. The applications to detecting poisoned and mislabeled data could have positive security implications.

**Claims And Evidence:**

Yes

**Requested Changes:**

(1) It would be good to add more detailed discussion of computational bottlenecks and potential solutions for scaling to larger graphs.

(2) It will be also great if authors could consider evaluation on additional graph types beyond citation and co-purchase networks, e.g. biological networks.

**Strengths And Weaknesses:**

Strengths:

(1) The study covers a wide range of data valuation methods and provides thorough experimental evaluation across multiple citation and co-purchase graph datasets with different GNN architectures. The experimental scope includes appropriate baselines with multiple evaluation criteria.

(2) The distinction between learning and overall signals is valuable for understanding how nodes influence both training and inference in transductive settings. The five key findings provide actionable insights about graph data valuation that extend beyond the i.i.d. setting.

(3) The paper includes solid experimental methodology with proper consideration of different signals, various utility functions, and extensive ablation studies.

Weaknesses:

(1) The dataset diversity is constrained to citation and co-purchase graphs. Evaluation on other graph types could give readers more comprehensive understanding and insights.

(2) The contribution is primarily empirical with limited technical novelty. The paper does not propose new data valuation methods specifically designed for graphs.

---

> ### Author Response · Authors · 2025-06-17
>
> We thank the reviewer for the thorough assessment and for highlighting the practical value of our *learning* vs. *overall* signal distinction.  Below we address each point in turn. Changes already incorporated in blue in the revised manuscript.
>
> **(W1/RQ2) Limited dataset diversity**
>
> Following your suggestion (and Reviewer BmWp’s), we now include two hyperlink-web heterophilic graphs, i.e. Texas and Wisconsin. We reported the new results in Appendix C.1. The heatmap values and the node removal plots mirror the trends on other datasets, reinforcing the generality of our five findings.
>
> **(RQ1) Computational bottlenecks and scalability**
>
> In Appendix D we break the total runtime into (i) *utility collection* and (ii) *value aggregation*. While we reported the sum of these times in the table, most of the computational cost comes from computing utilities on the subsets. To scale to larger graphs, we employ the SGC linearized version which allows to compute the weights in closed form (see Appendix D). Adopting this model, we obtain a ~$3\times$ speed up over GCN which is trained using stochastic gradient descent for approaches like DM, $\alpha$-Banz and data Shapley. Additionally, to scale further, we need either a procedure that requires no training (such as influence functions, though these have shown poor performance in highly non-convex settings), or one that approximates true values using minimal subsets to reduce the utility collection cost. As highlighted in the paper, we think that maximum sample reuse is a crucial feature for (graph-based) data valuation approaches, since reusing subsets for computing multiple (node) values enables better approximation for the same number of subsets. Following the suggestion, we have expanded on MSR as a scaling solution in the Discussion paragraph of Section 4.
>
> **(W2) Limited technical novelty**
>
> Our goal was to establish a solid baseline for graph valuation before jumping into proposing graph-specific valuation techniques and a roadmap for future development (e.g. we find that sample reuse is key). Indeed, our findings show that generic techniques (α-Banzhaf and Datamodels) outperform graph-specific techniques (PCW). Moreover, we have conceptual contributions in terms of four orthogonal graph-specific valuation settings (*learning vs. overall* × *train vs. all*).

---

### Decision · Action_Editor_BcaR · 2025-07-10

**Recommendation:** Accept with minor revision

**Additional Comments:**

I have a few minor comments regarding the presentation that require the authors’ attention:

1. The placement of footnote numbers should follow standard formatting conventions. Specifically, the footnote in the abstract should appear after the period. The second footnote is correctly placed.
2. The experimental setting lacks clarity, as also noted by another reviewer. While Section 4 begins with a description, it primarily focuses on datasets and baseline methods. More details on the experimental procedures and evaluation metrics are needed.
3. Figure captions could be made clearer. Many figures include multiple subfigures, and the relationships among them should be explained within the caption.

**Audience:**

Yes

**Audience Explanation:**

Data valuation is a new direction, which catches increasing attention in the machine learning community.

**Claims And Evidence:**

Yes

**Claims Explanation:**

This is an empirical study on graph data valuation, where the authors provide 5 key findings by conducting extensive experiments across multiple datasets and methods. The experimental design is sound and the results provide valuable insights.